# Transformer brain encoders explain human high-level visual responses

**Hossein Adeli**
Zuckerman Mind Brain Behavior Institute
Columbia University
ha2366@columbia.edu

**Sun Minni**
Zuckerman Mind Brain Behavior Institute
Columbia University
ms5724@columbia.edu

**Nikolaus Kriegeskorte**
Zuckerman Mind Brain Behavior Institute
Columbia University
nk2765@columbia.edu

## Abstract

A major goal of neuroscience is to understand brain computations during visual processing in naturalistic settings. A dominant approach is to use image-computable deep neural networks trained with different task objectives as a basis for linear encoding models. However, in addition to requiring estimation of a large number of linear encoding parameters, this approach ignores the structure of the feature maps both in the brain and the models. Recently proposed alternatives factor the linear mapping into separate sets of spatial and feature weights, thus finding static receptive fields for units, which is appropriate only for early visual areas. In this work, we employ the attention mechanism used in the transformer architecture to study how retinotopic visual features can be dynamically routed to category-selective areas in high-level visual processing. We show that this computational motif is significantly more powerful than alternative methods in predicting brain activity during natural scene viewing, across different feature basis models and modalities. We also show that this approach is inherently more interpretable as the attention-routing signals for different high-level categorical areas can be easily visualized for any input image. Given its high performance at predicting brain responses to novel images, the model deserves consideration as a candidate mechanistic model of how visual information from retinotopic maps is routed in the human brain based on the relevance of the input content to different category-selective regions. Our code is available at https://github.com/Hosseinadeli/transformer_brain_encoder/.

## 1 Introduction

An influential approach to study plausible neural computations in the brain is to train Deep Neural Network (DNN) models on different tasks [44, 28] and compare their learned representation to brain activity [52, 23]. There has been a great deal of discussion and research on best ways to compare the learned representations to the ones recorded from the brain (across models and across models and brains). One main approach is to build encoding models— learn a mapping function from one feature domain to another and measure the accuracy of the prediction in held-out sets [14, 36]. An alternative approach is to characterize the geometry or topology of the representation in each model or in the brain and then compare them (e.g. RSA; [29]). In this work, we focus on the learned encoding functions, as we believe that it can give us further insight into the computations in the brain.

39th Conference on Neural Information Processing Systems (NeurIPS 2025).

The visual system uses structured retinotopic maps as it processes visual information in the cortex. Not surprisingly, models, such as Convolutional and transformer neural networks, that also maintain retinotopic maps of the space perform best on different visual tasks (e.g. recognition and segmentation) and consistently outperform other models in different brain activation prediction benchmarks [46, 17]. However the retinotopic feature maps from deep networks presents typically have a very large number of units posing us with a challenge when mapped unto the responses in the brain. Linear encoding models, although theoretically the simplest choice, can become very high-dimensional in that case (the number of parameters equals the product of the number of model units and the number of units/voxels to be predicted) and require strong regularization (L2 penalty) given the size of typical neuroimaging datasets [36]. To address these limitations, approaches have been proposed that learn spatial receptive fields (RF) for different units or voxels in the brain data, using which the representation is first aggregated across space and then the lower dimensional representation is linearly mapped to the brain responses [26, 49, 35]. These models have been shown to perform on par with linear regression models despite having a fraction of the number of parameters and are also more plausible mechanisms of how information can route to different units. However, they can only capture fixed routing where input to a unit comes from a specific area in space regardless of the input content.

Transformer architectures have been extremely successful in many domains, including vision [12] and language [51]. Their success can be attributed to a general and simple (therefore scalable) computational motif where information is routed based on the content. In these models, each token (be a representation of a word in a sentence or a patch in an image) queries other tokens to find how relevant they are to updating its representation. The selective nature of this mixing has motivated naming this process "attention" in Transformers [51]. Then the new representation of this token becomes the average of the representation of all tokens, weighted by their degree of relevance (i.e. attention scores). We hypothesize that the optimal way for the routing of information from the retinotopic visual maps to category selective areas is to use the same computational motif where brain areas only attend to parts of the visual maps with the content relevant to what the area is selective for (Fig. 1). For example if there is face in the image, it could appear anywhere, but the FFA (fuisform face area) can learn to route only the information from the patches where the face-like stimuli are and then expand this lower dimensional representation in the area. Note that this approach is in a way a generalization of the aforementioned RF based methods going from fixed receptive fields to a dynamic content-based receptive fields.

## 2 Related works

**Brain encoding models:**   Predicting brain activity is an important objective, both as an engineering challenge and also as a means of studying brain computations, reflected in the number of community-driven benchmarks such as Algonauts [17], Brain-score [46], and Sensorium [50]. The availability of large-scale neural datasets has necessitated innovation in new encoding models [21]. Spatial-feature decomposition models have shown that considering the retinotopic maps and the receptive field organization can lead to more efficient encoding models [26, 49, 35, 45]. Generalizing these approaches to high-level visual areas would require considering more dynamic routing motifs.

**Self-supervised Vision Transformers:**   Transformers have been shown to outperform convolutional and recurrent neural networks (CNNs) on a variety of visual tasks including object recognition [12]. More recent studies have explored training these models on self-supervised objectives, yielding some intriguing object-centric properties [1] that are not as prominent in the models trained for classification. When trained with self-distillation loss (DINO, [6] and DINOv2 [38]), the attention values contain explicit information about the semantic segmentation of the foreground objects and their parts, reflecting that these models can capture object-centric representations without labels [1]. These findings show that features from these models can be a good basis for predicting neural activity in the brain. Recent work has also shown that networks trained using self-supervised contrastive losses (such as SimCLR; [9]) match the predictive power of supervised models for high-level ventral-stream visual representations in the brain [27, 8]. These works argue for self-supervised learning methods as a more plausible objective function for learning brain like visual representations.

**Encoder-decoder Vision Transformers:**   Transformer-based encoder-decoder models provide a general framework that has achieved great performance in many domains [51] including domains

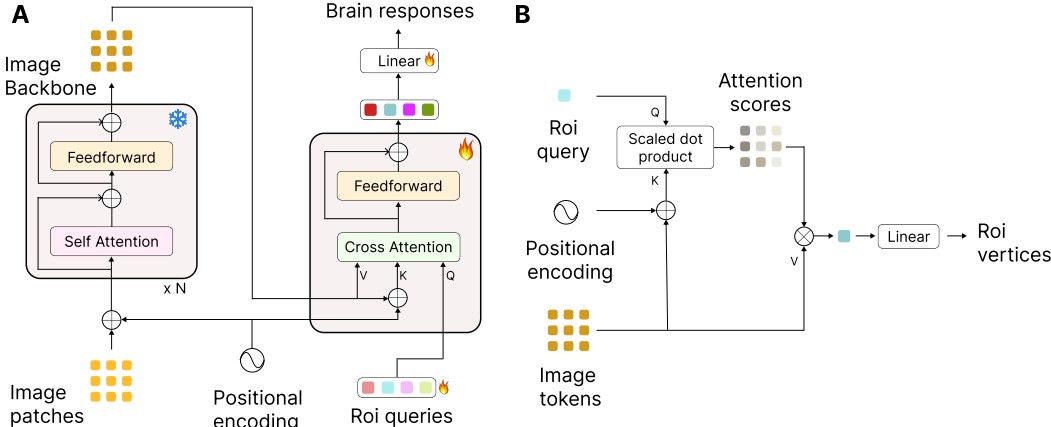

Figure 1: **A. T**ransformer **B**rain **En**coder (TBEn) architecture. The input patches are first encoded using a frozen backbone model. The features are then mapped using a transformer decoder to brain responses. **B.** The cross attention mechanism showing how learned queries for each ROI can route only the relevant tokens to predict the vertices in the corresponding ROI.

where one modality (e.g. image) is mapped onto another one (e.g. language) [41]. A related pioneering work to our approach is the DETR model [5] applied to the problem of object detection and grouping in images. The encoder in this model converts the image to rich object-centric features. The decoder uses learnable embeddings, called queries, corresponding to different potential objects, that gather information from the encoder features using cross-attention over several layers. After the decoding process, each object query can then be linearly mapped into to the category and bounding box for an object. The model is trained end-to-end and can detect many objects in one feedforward pass. We also employ this general framework here.

## 3 Methods

### 3.1 Dataset

We run our experiments on the Natural Scene Dataset (NSD; [3]) where the fMRI (functional magnetic resonance imaging) responses were collected from 8 subjects, each seeing up to 10,000 images. The reported results are from subjects 1, 2, 5, and 7 who completed all recording sessions. The surface-based fMRI responses across the three repetitions of each image were averaged for model training and testing. We use the train/test split that was introduced in the Algonauts benchmark [17] where the last three sessions for each subject were held out to ensure that no test data were accessed during the model development and to make the prediction task as natural as possible (predicting the future responses). All models were trained to predict the most visually responsive vertices in the brain [1]. Our analyses focused on as subset of approximately 15k vertices for each left and right hemispheres (LH and RH), shown in Figure 2A on a surface map. ROI level labels were provided for all the selected vertices based on visual and categorical properties (using auxiliary experiment; refer to [3] for details). The labels are for early visual areas ('V1v', 'V1d', 'V2v', 'V2d', 'V3v', 'V3d', and 'hV4'), body selective areas ('EBA', 'FBA-1', 'FBA-2', and 'mTL-bodies'), face selective areas ('OFA', 'FFA-1', 'FFA-2', 'mTL-faces', and 'aTL-faces'), place selective areas ('OPA', 'PPA', 'RSC'), and word selective areas ('OWFA', 'VWFA-1', 'VWFA-2', 'mfs-words', and'mTL-words').

### 3.2 Transformer Brain Encoder (TBEn)

We apply the general transformer encoder-decoder framework to map images to fMRI responses. Figure 1A shows the architecture of our model. The input image is first divided into patches ($31 \times 31$

---

[1]In fMRI, a vertex refers to a point on the surface mesh of the cortex used in surface-based analysis. It is analogous to a "voxel" (volumetric pixel) in volumetric fMRI data, but defined in the 2D cortical surface space.

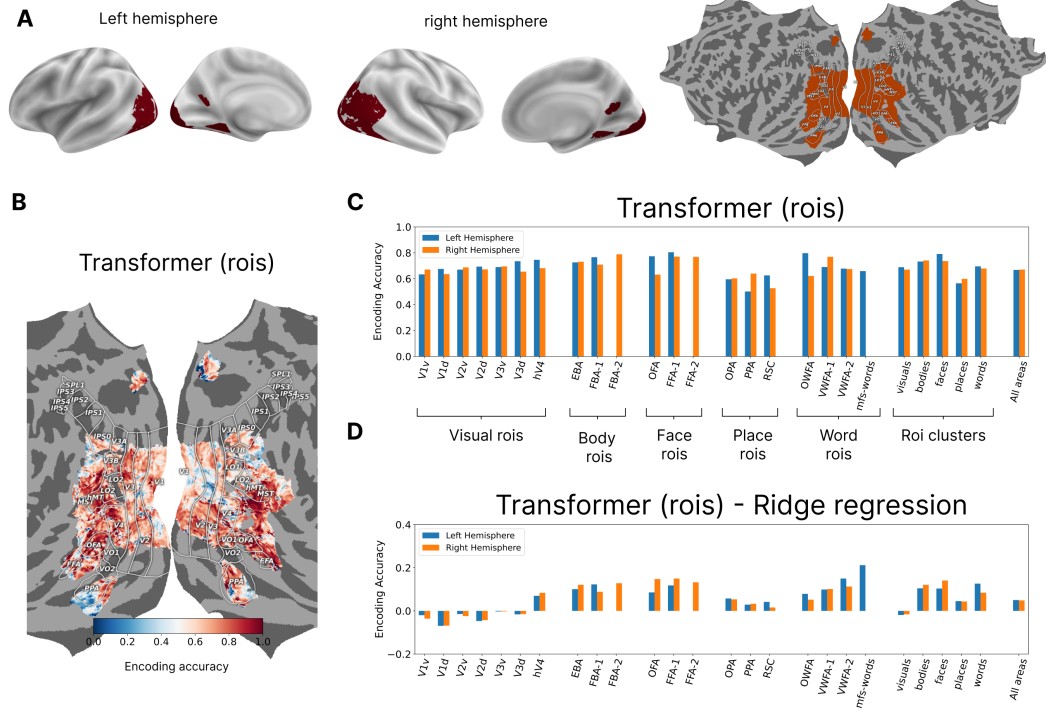

Figure 2: **A.** The general region of interest for highly visually responsive vertices in the back of the brain shown on different surface maps. **B.** Encoding accuracy (fraction of explained variance) shown for Subject 1 for all the vertices for the transformer model using ROIs for decoder queries. **C.** Encoding accuracy for individual ROIs and for ROI clusters based on category selectivity for the two hemispheres. **D.** The differences in encoding accuracy between the transformer and the ridge regression models showing that improvement in the former is driven by better prediction of higher visual areas.

in our dataset) of size $14 \times 14$ pixels. These image patches are input to the backbone model which is a 12-layer vision transformer and frozen to be used as a feature backbone.

The decoder uses input queries corresponding to different brain ROIs in different hemispheres to gather relevant information from the backbone outputs for predicting neural activity in each ROI. Note that these queries are learnable embeddings for each ROI trained as part of the model training. We use a single-layer transformer for the decoder with one cross-attention and a feedforward projection operation. Figure 1B shows the cross-attention process. The positional encoding is added to the image token representation to create the keys. This allows the ROI query to attend either to the location or the content of the input tokens through scaled dot-product attention. The attention scores are then used to aggregate all the image tokens that are relevant to predict the brain activity in that ROI. The output decoder tokens are then mapped using a single linear layer to fMRI responses of the corresponding ROI. In our implementation, decoder output for each ROI is linearly mapped to a vector with the size equal to the number of vertices in that hemisphere. The response is then multiplied by a mask that is zero everywhere except for the vertices belonging to that ROI. This masking operation ensures that the gradient signal feeding back from the loss will only train linear mappings to the vertices of the queried ROI. The responses from different ROI readouts will then be combined using the same masks to generate the prediction for each hemisphere. The ROI queries, transformer decoder layer and the linear mappings are trained with the Adam optimizer [25] using mean-squared-error loss between the prediction and the ground truth fMRI activity for each image. We train and test the models separately for each subject.

# 4 Experiments

For all models including all the baselines, we did 10-fold cross validation using the training set for each subject and averaged the model predictions across all folds. The model predictions were evaluated first using Pearson correlation between the predictions and the ground truth test data. The squared correlation coefficient were then divided by the noise ceiling (see [3] Methods, Noise ceiling estimation) to calculate the encoding accuracy as the fraction of the explained variance.

We present results using multiple different feature backbones namely, DINOv2 base model [38], ResNet50 [20], and CLIP large model [41]. For the DINOv2 backbone, inspired by prior work on human attention prediction [1], we did some preliminary analyses and found the patch level query representations (instead of values) to have slightly more predictive power and chose to use them in all our experiment. For ResNet50, the feature maps from the last layer were extracted and reshaped to create the visual tokens comparable to transformers. For CLIP, we chose the large model to have the same image patch size (14) and transformer token dimension (768) to the DINOv2 base model. Unless otherwise stated, the features from the last layer of the backbone models are used as the input representation to the decoder.

We consider multiple different mapping functions to compare to our proposed method. The Ridge regression model flattens the feature representation across space and feature dimensions and learns one linear mapping to the fMRI responses. We used a grid search to select the best ridge penalty to maximize performance on the validation data. The CLS + regression model linearly maps only the CLS token from the transformer backbone to the vertex responses. This is a common practice in many neuroscience studies to make the number of parameters more tractable. Another common model is to first reduce the dimensionality of the features using Principle Component Analyses (PCA) and then learn the linear mapping to the brain responses (PCA + regression).

For the spatial-feature factorized method, the model learns a (H × W) spatial map and applies that to the input feature similar to the attention map in Figure 1B. The scores however are only learned for a given ROI or a vertex and are not dependent on the content of the image. The spatial map then aggregates the features to be linearly mapped to the brain responses. The Saliency based integration method uses saliency map of the image, instead of a learned spatial map, to integrate the tokens across space [24]. To implement this baseline, we used DeepGaze [30], a state of the art saliency model, to generate bottom-up saliency maps for each image and then resized the maps and used the resulting attention values (weights) to combine the token representations to create a single token. A linear regression was then trained to map these compressed representations to vertex activations.

For the transformer brain encoder, we used 24 queries per hemisphere corresponding to the 24 ROIs. Note that not all ROIs were present in all the subjects, therefore we present results and figures for subjects individually. If an ROI is not mapped in a subject the decoder output is not mapped to any vertices. The figures in the main text are generated using the results from subject 1, but the figures for the remaining three subjects are presented in the supplementary section A.1.

Table 1: Encoding accuracy using DINOv2 backbone

| Encoder | Subjects | | | | Model size (M) |
|---|---|---|---|---|---|
| | S1 | S2 | S5 | S7 | |
| Ridge regression | 0.56 | 0.52 | 0.50 | 0.37 | ~900 |
| CLS + regression | 0.38 | 0.37 | 0.45 | 0.33 | ~22 |
| PCA + regression | 0.52 | 0.47 | 0.46 | 0.34 | ~22 |
| Spatial-feature factorized (rois) | 0.49 | 0.46 | 0.48 | 0.37 | ~23 |
| Saliency based integration | 0.38 | 0.37 | 0.44 | 0.32 | ~22 |
| Transformer (rois) | **0.60** | **0.56** | **0.56** | **0.42** | ~28 |

Table 1 shows the encoding accuracy of the encoding models using the DINOv2 backbone. Ridge regression requires tuning a larger number of parameters compared to the other approaches (all model sizes reported as multiples of millions of parameters).

The CLS token takes a weighted average of all the salient image tokens to create a compact representation of the image and the linear mapping of this token performs similarly to the Saliency based integration model. Both serve as useful comparisons to highlight the benefit of transformer attention

versus generic feature reweighting. The PCA based model performs similarly to the Spatial-feature factorized but both perform worse than the full ridge regression model. Our model, leveraging the attention mechanism to flexibly route information [2, 4, 40], consistently outperforms all the baseline models across all subjects. The important difference to note is that our model allows each ROI to dynamically route the tokens that have relevant content for that ROI so in other words each learn to create their own "CLS" token dynamically based on the content of the image and the ROI selectivity.

Figure 2B shows the encoding accuracy of our model for subject 1 for the areas of interest projected onto the cortical surface using Pycortex [15]. Figure 2C shows the encoding accuracy divided over all the individual ROIs and also clusters of ROIs. When we compare the transformer encoder to the ridge regression model (Fig. 2D), we see that our model achieves higher encoding accuracies through better performance for categorical areas. This suggests that content based routing can be part of the brain computation for higher level visual areas.

We further tested whether the transformer based mapping requires a larger number of training images to be effective. Table 2 shows that the model can be trained using as little as a few hundred samples making it suitable for smaller scale experiments as well. Also encouraging is to see that our model can achieve accuracy on par with the baseline models with a fraction of the training data.

Table 2: Encoding accuracy for different training set sizes

| Encoder | Training set size | Subjects | | | |
|---|---|---|---|---|---|
| | | S1 | S2 | S5 | S7 |
| Transformer (rois) | 550 | 0.41 | 0.45 | 0.45 | 0.34 |
| Transformer (rois) | 1100 | 0.46 | 0.48 | 0.49 | 0.36 |
| Transformer (rois) | 2200 | 0.51 | 0.52 | 0.51 | 0.39 |
| Transformer (rois) | 4400 | 0.56 | 0.54 | 0.54 | 0.40 |
| Transformer (rois) | 8800 | **0.60** | **0.56** | **0.56** | **0.42** |

To examine whether our results depend on the specific choice of the transformer backbone architecture, we tested all the encoding models on the ResNet50 backbone features (a fully convolutional network). Table 3 shows that we replicate the exact same pattern of accuracy as the DINOv2 backbone, where the transformer encoder outperforms the other two alternatives across all subjects. This shows that the transformer encoder can map differently learned features (transformer vs convolution) well to the brain data.

Table 3: Encoding accuracy using ResNet50 backbone

| Encoder | Subjects | | | | Model size (M) |
|---|---|---|---|---|---|
| | S1 | S2 | S5 | S7 | |
| Ridge regression | 0.49 | 0.48 | 0.47 | 0.37 | ∼900 |
| Spatial-feature factorized (rois) | 0.42 | 0.42 | 0.43 | 0.33 | ∼60 |
| Transformer (rois) | **0.52** | **0.50** | **0.50** | **0.38** | ∼28 |

## 4.1 Vertex-based routing

So far the presented transformer encoding models used ROIs as units of routing. But the routing could be made more granular by learning a decoder query for each vertex where the gathered features from the decoder would be mapped linearly to the corresponding vertex value. This approach can also be applied in the spatial-feature encoding models where a spatial map is learned per vertex. Table 4 shows model accuracies for these two approaches using the vertex-based routing, indicating improvements for both models across all the subjects. Examining the encoding accuracy for individual ROIs (Fig. 3), we can see that the performance boost came almost entirely from early visuals areas for the transformer based model. The fact that shifting from ROI-based to vertex-bases routing does not improve encoding accuracy for higher visual areas indicates that ROIs may be the right level of routing for those regions, however the early visual areas requires more granular routing because the receptive fields of the vertices are smaller, more heterogeneous, and less content dependent.

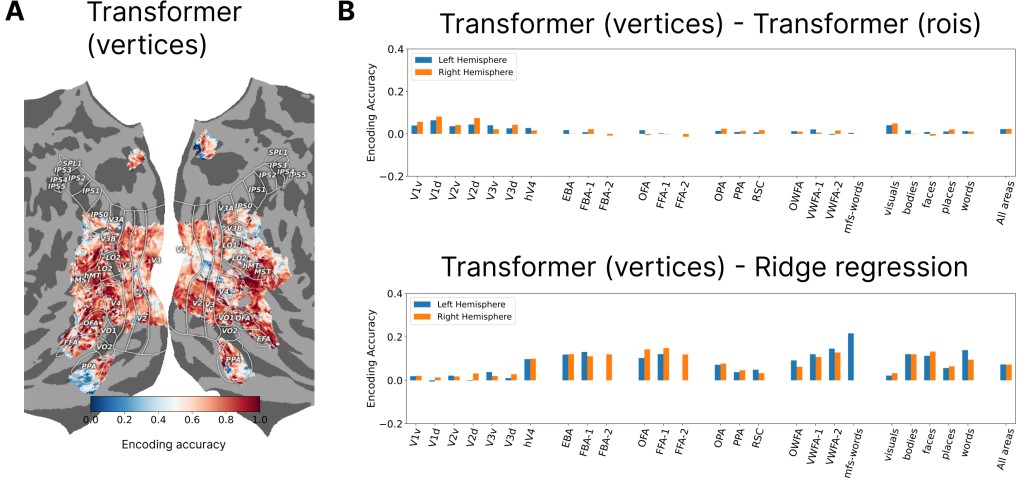

Figure 3: **A.** The encoding accuracy for subject 1 shown on the brain surface for the transformer model with vertices as decoder queries. **B.** The difference in encoding accuracies going from ROIs to vertices as the decoder queries shows the improvement is almost entirely from the early visual areas. **C.** The vertex-based transformer model outperforms the ridge regression model for almost all the ROIs.

Comparing the vertex-based transformer model to the ridge regression model (Fig. 3B) shows that the former now outperforms the latter in almost all the ROIs.

Table 4: Encoding accuracy for different decoder queries

| Encoder | Subjects | | | | Model size (M) |
|---------|----|----|----|----|----------------|
| | S1 | S2 | S5 | S7 | |
| Spatial-feature factorized (rois) | 0.49 | 0.46 | 0.48 | 0.37 | ~23 |
| Spatial-feature factorized (vertices) | 0.52 | 0.48 | 0.48 | 0.37 | ~51 |
| Transformer (rois) | 0.60 | 0.56 | 0.56 | 0.42 | ~28 |
| Transformer (vertices) | **0.63** | **0.59** | **0.57** | **0.44** | ~50 |
| Transformer (vertices) backbone layers ensemble | **0.65** | **0.62** | **0.59** | **0.45** | ~300 |

Motivated by previous encoding models of the brain having used CLIP embeddings [41] to represent images [32], we tested the different mapping functions using this feature backbone. Table 5 shows while the performance is generally not as good as the DINOv2 backbone, it yields the same exact pattern of results. The Transformer-based models outperform other alternatives with the vertex-based routing reaching higher performance overall. Taken together with also the lower performance we saw with ResNet50 backbone, the DINOv2 features, a self-supervised trained vision transformer, deserve consideration as models of human visual brain representations.

Table 5: Encoding accuracy using CLIP vision backbone

| Encoder | Subjects | | | | Model size (M) |
|---------|----|----|----|----|----------------|
| | S1 | S2 | S5 | S7 | |
| Ridge regression | 0.51 | 0.48 | 0.47 | 0.38 | ~490 |
| Spatial-feature factorized (rois) | 0.38 | 0.35 | 0.40 | 0.31 | ~22 |
| Spatial-feature factorized (vertices) | 0.44 | 0.40 | 0.42 | 0.32 | ~30 |
| Transformer (rois) | 0.53 | 0.49 | 0.50 | 0.38 | ~28 |
| Transformer (vertices) | **0.55** | **0.52** | **0.52** | **0.40** | ~50 |

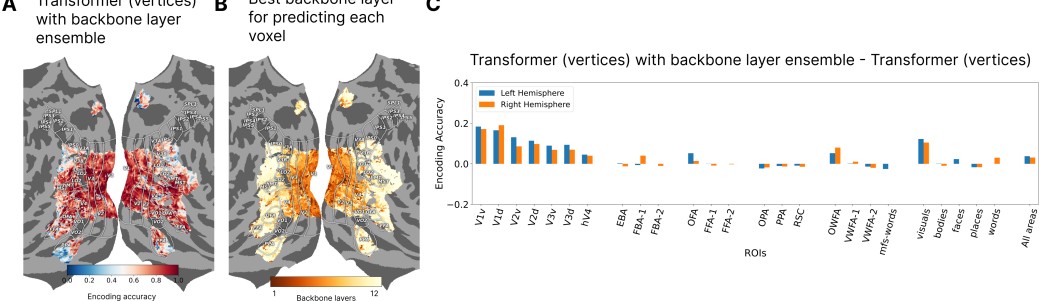

Figure 4: **A.** Encoding accuracy of the transformer encoding model with vertex-based queries ensembled across backbone layers. **B.** Showing the backbone layer from which each vertex was best predicted. **C.** The improved performance of ensembling is almost entirely from better prediction of early visual areas.

## 4.2 Ensemble

A concern with using complex encoding models for neural system identification is that the non-linear mapping may obscure the differences in the underlying representations [21]. However, our results with different feature backbones show that the ones that perform better using the linear model consistently perform better using our transformer encoding model as well, just with the latter achieving higher accuracies.

To address this concern further, we consider a robust phenomenon shown consistently using linear encoding with convolutional neural network backbones, where the earlier layers of the network are better features for predicting the earlier visual areas [52, 19, 37, 23, 53]. We trained different transformer decoders with image tokens coming from different layers of the DINOv2 backbone. We then use a softmax operation across the ensemble of models to get the final prediction for each voxel. The softmax weights are based on goodness of the prediction for each model for that vertex in the validation set. Figure. 4A shows the accuracy of the overall model on the brain surface for subject 1. The layers that had the highest weights in the ensemble for predicting for each given voxel is shown in 4B; higher visual areas were better predicted by later backbone layers, indicating that backbone layers capture similar feature abstractions as the brain.

Comparing the ensemble model to the model trained using only the final backbone layer features (Fig. 4C), we can see that the performance increase is entirely driven by better prediction of earlier visual areas. These results show that our encoding model does not obscure the differences in the underlying representation pointing further to its plausibility.

## 4.3 Attention maps

Different methods have been developed to interpret linear encoding models to make claims about the the selectivity learned for each ROI. Some methods tend to retrieve or generate images that highly activate the ROI vertices [33, 34, 7], and others focus on creating importance maps to show which parts of the input images are important for predicting the activity of an ROI [43].

The difference in our approach is that the cross-attention scores (Fig. 1B) can be examined to reveal the selectivity for each ROI, making our model inherently more interpretable. We visualize the attention maps for 3 different ROIs in Figure 5 for the transformer encoder trained with ROI decoder queries with DINOv2 backbone. First is an early visual area, V2d (dorsal) in the left hemisphere. Since the visual field is flipped around both horizontal and vertical meridians in the cortex (starting from the retina), we expect the brain activity in this area to represent visual information from the bottom-right of the input (given that the subjects were instructed to hold fixation at the center of the screen for the presentation duration). We see this exact pattern emerge in the attention maps. Recall that the decoder queries can learn to attend to both patch locations or their content (since the key value is the sum of backbone image patches and positional encoding). In this case, the attention seems to be completely driven by the location, similarly for all the images, ignoring the content. This

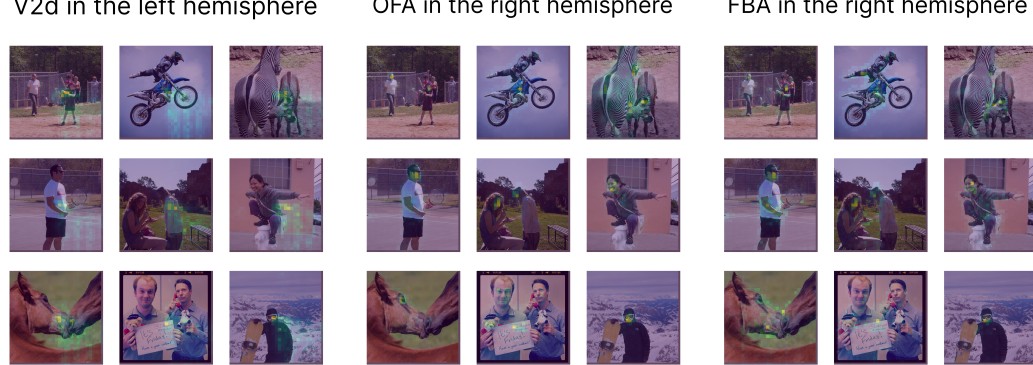

| V2d in the left hemisphere | OFA in the right hemisphere | FBA in the right hemisphere |

Figure 5: **Attention maps.** Transformer decoder cross attention scores for three ROIs overlaid on the images (with brighter colors indicating higher attention weights). The selected ROIs show different ways in which the learned ROI queries can route information— based on location (V2d), content (FBA), or a combination of the two (OFA) depending on the location of the ROI in the brain processing hierarchy.

is exactly what we would expect from an early visual area. The fact that all the vertices in this ROI have to share the same attention map hurts accuracy as we saw in Figure 2D, since the vertices do have smaller RFs in this area than a quadrant, however this can be addressed by vertex level routing.

The second ROI is OFA in the right hemisphere, a mid-level face selective area [16]. The attention maps is this area consistently focus on faces. Since this area is in the right hemisphere it also has a preference for visual input in the left visual field. We can see this for cases with multiple faces where the faces in the right visual field are not strongly attended. The decoder query therefore makes use of both the positional encoding and the content component of the key to attend to the most relevant part of the image to predict vertices in this ROI. The attention could also be spread across multiple faces in different locations. This is the important dynamic aspect of the receptive field in higher visual areas that can be captured using the transformer attention mechanism. The third area is FBA in the right hemisphere, a high level body selective area [39]. The attention maps are more spread across bodies for this ROI and not just faces. Supplementary section A.2 includes a quantitative analyses of the category selectivity of the attention maps. In the Supplementary section A.3, we provide an analyses of the similarity between the learned queries for different ROIs (capturing visual and semantic similarity between them) and also show how our model can be used in a pipeline using diffusion models [32] to generate stimuli that maximally activate different ROIs (section A.4).

Table 6: Encoding accuracy using BERT backbone

| Encoder | Subjects | | | | Model size (M) |
|---|---|---|---|---|---|
| | S1 | S2 | S5 | S7 | |
| Ridge regression | 0.19 | 0.21 | 0.25 | 0.19 | ∼900 |
| Transformer (rois) | **0.27** | **0.27** | **0.33** | **0.27** | ∼28 |

### 4.4 Text modality

So far we have tested the transformer encoding model on a few vision backbones but is this approach generalizable to other modalities? To test this, we first used the BLIP model [31] to generate short captions for all the images in the dataset. Using BERT [10] as the feature backbone, the decoder works exactly as before, using ROI queries to map backbone features to fMRI responses. Table 6 shows how the transformer model outperforms the regression model across all subjects (with a fraction of the parameters). Given only semantic information available in the captions, the model can only predict the high level visual areas as shown in Figures 6A and 6B.

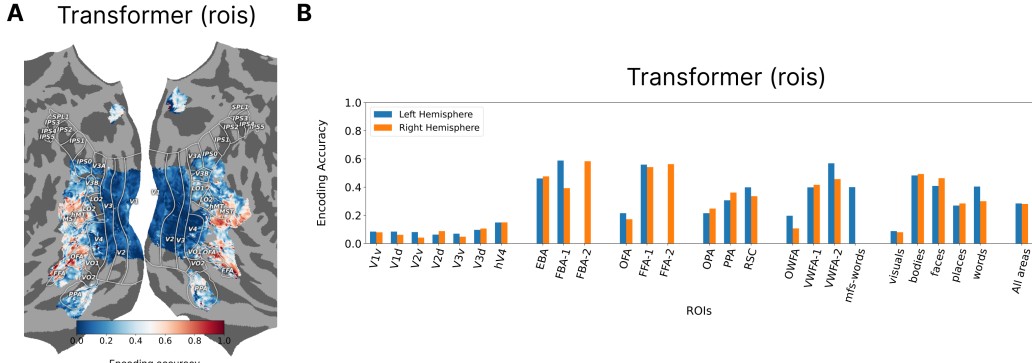

Figure 6: **A.** Transformer encoder accuracy using image caption as input **B.** Only high-level visual areas are predicted by semantic information in a caption.

## 5 Discussion

Linear encoding models have been the dominant method used for learning the mapping from model features to brain activity [14]. The reasons for this (see [21] for a review of these points) include theoretical simplicity, allowing comparison among backbone features, biological plausibility, and the ability to interpret the learned weights. However, this approach is parameter inefficient for a typical number of voxels and image features, ignores the organization of the features, and does not capture nonlinear computations between brain areas such as ubiquitous normalizations [48]. Our proposed routing based method not only reaches state of the art accuracy, it also achieves the aforementioned desiderata for encoding models, as we have shown in our results.

Foundation vision models (e.g. DONOv2 or CLIP) trained with self-supervised objectives can serve as general visual representation backbones. However these task agnostic models do not capture all the computations in the brain and between brain areas, which needs to be addressed by learning better encoding models. Our work suggests a mechanism for how different brain areas dynamically gate their input based on the input content and the area selectivity. A flexible routing mechanism is reflected in deviation from the classical RF characterization of responses, so content-dependent RF shifts provide evidence for a more flexible mechanism [42, 18]. Our results showing that the encoding accuracy for high-level areas cannot be improved beyond ROI-based routing also agrees with prior work on between area interactions using communication subspaces [47] which can also be modulated by attention. The routed information that is relevant to an area can then get expanded more in-depth. This process allows for cutting down on wiring cost in the brain by not connecting all the units in one area to another area but rather only a subset of relevant information getting routed with more local connections expanding the representation.

**Limitations:** We performed our experiments on NSD [3], the largest image viewing fMRI dataset to date. It will be important to test the generality of our approach on other datasets using different recording techniques (Neurophysiology, EEG, etc) [13] and on different input modalities (such as video and audio). We used vertex-wise routing to capture the responses in early visual areas but while the computations for smaller receptive fields can be learned by this approach, the way the RFs are implemented in the brain are through different anatomical and wiring constraints. Also we chose for the model to read out the brain responses from a backbone for both early and high-level visual areas. Future work will seek to explore the connectivity between early and high-level visual areas in a more integrated system and test whether making the model further aligned with known anatomy of the visual cortex will improve performance.

## Acknowledgments

Research reported in this publication was supported in part by the National Institute of Neurological Disorders and Stroke of the National Institutes of Health under award numbers 1RF1NS128897 and 4R01NS128897. The content is solely the responsibility of the authors and does not necessarily represent the official views of the National Institutes of Health.

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

# A    Supplementary Material

## A.1    Encoding accuracies for Subjects 2, 3 and 7

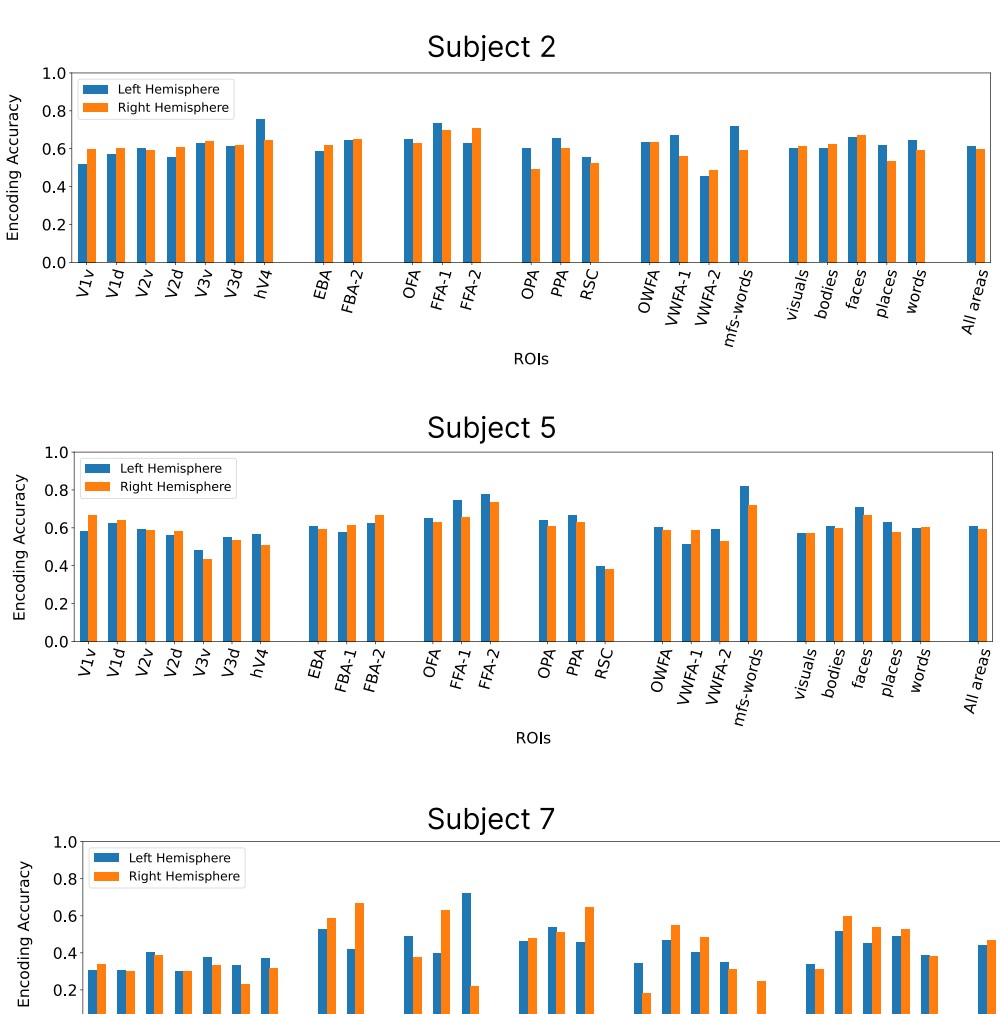

Figure S1: Encoding accuracy (fraction of explained variance) shown for Subjects 2, 5, and 7 for individual ROIs and for ROI clusters for the two hemispheres. The transformer model uses ROIs for decoder queries and features from the last layer of the DINOv2 backbone.

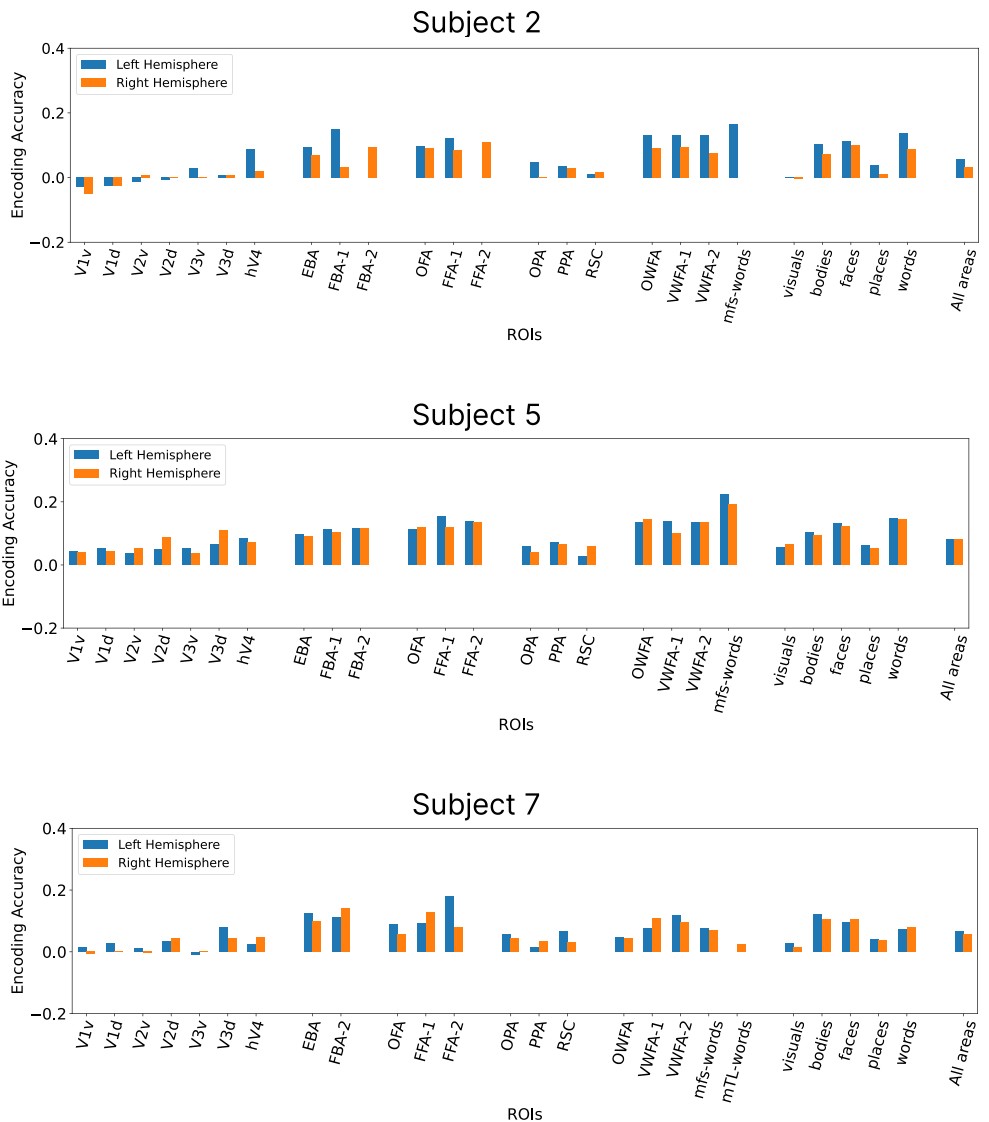

Figure S2: The differences in encoding accuracy between the transformer and the ridge regression models shows that the transformer encoder better predicts especially higher visual areas.

# Transformer (vertices)

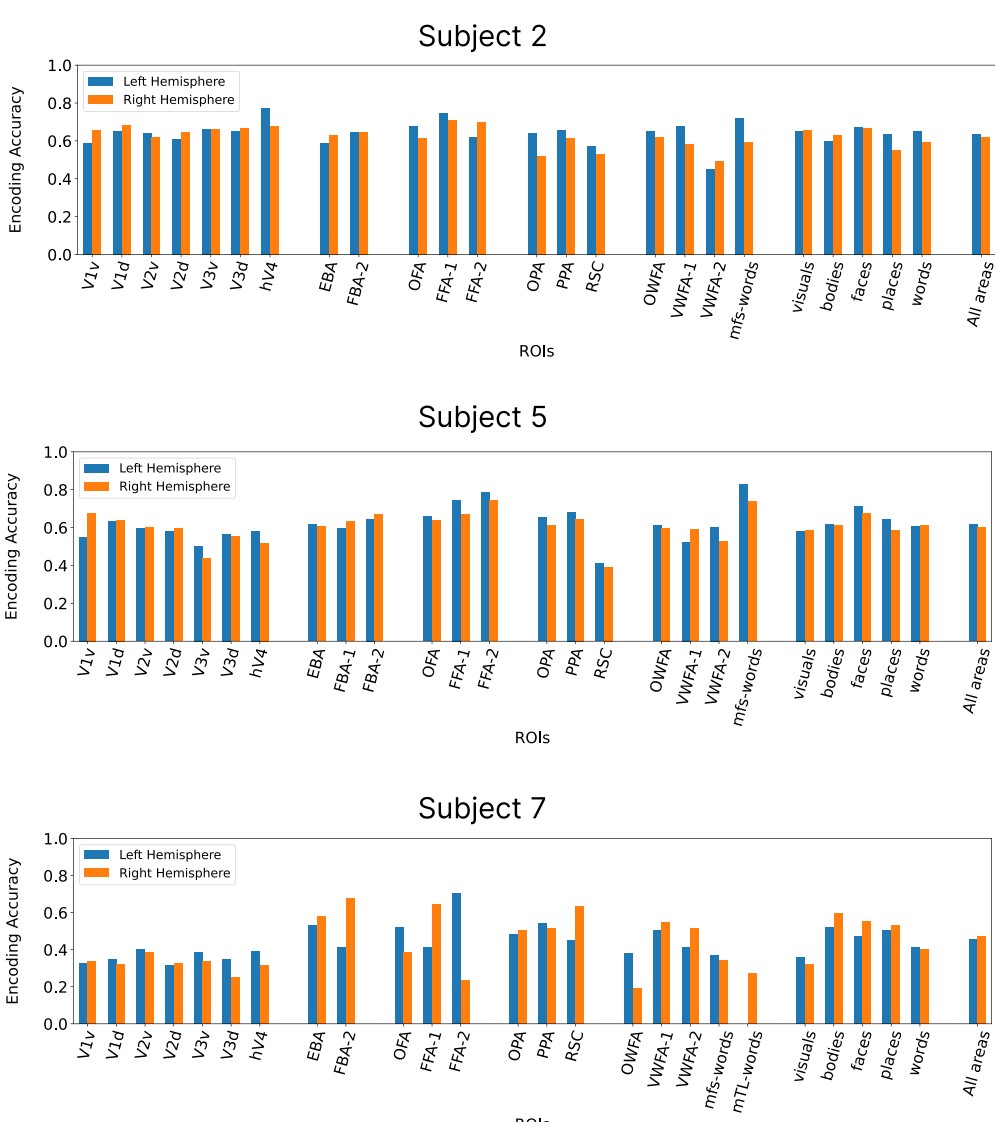

Figure S3: Encoding accuracy (fraction of explained variance) shown for Subjects 2, 5, and 7 for individual ROIs and for ROI clusters for the two hemispheres. The transformer model uses vertices for decoder queries and features from the last layer of the DINOv2 backbone.

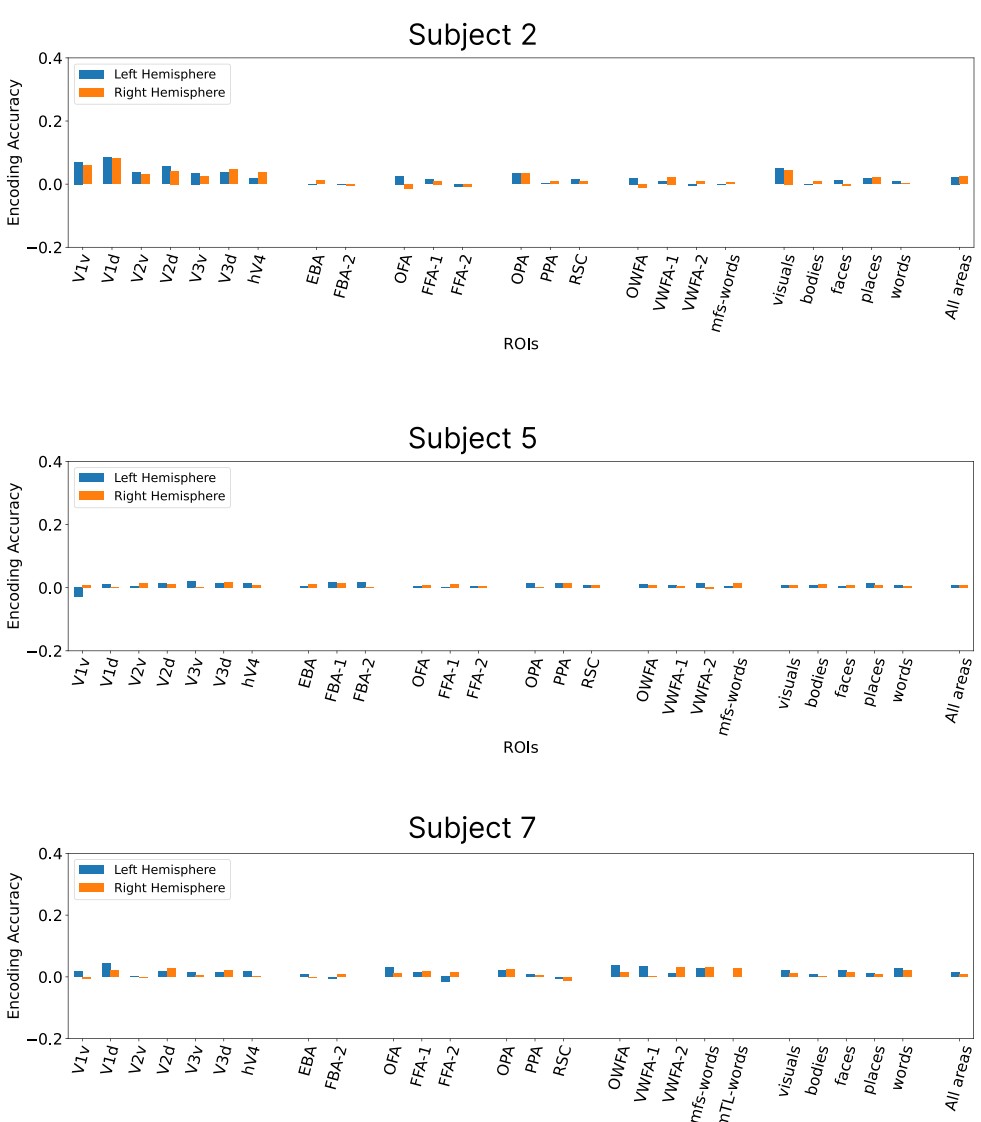

Figure S4: The differences in encoding accuracy between the transformer model using vertices and the model using ROIs as decoder queries. The figure shows that any potential improvement in the former is driven by better prediction of early visual areas.

## A.2 Category selectivity of attention maps

To quantify the category selectivity of attention maps, we classified each pixel of the test set images using YOLOv5 [22] and YOLOv8-face [11] into five categories: background, face, body (pixels classified as person but not as face), animal, and food. For each ROI, we calculated and resized its attention maps to $434 \times 434$ for these images, and reported the categories of the 2k pixels with top attention values. We found that the category selectivity is consistent with ROI labels, with EBA most selective for body, FFA most selective for face, and OPA/PPA/RSC most selective for background.

Table 7: Category selectivity of ROI attention for subject 1.

|       | background | face | body | animal | food |
|-------|------------|------|------|--------|------|
| EBA   | 0.03       | 0.36 | 0.61 | 0.00   | 0.00 |
| FFA-1 | 0.00       | 0.79 | 0.16 | 0.05   | 0.00 |
| FFA-2 | 0.00       | 0.83 | 0.17 | 0.00   | 0.00 |
| OPA   | 0.54       | 0.12 | 0.14 | 0.05   | 0.15 |
| PPA   | 0.44       | 0.25 | 0.11 | 0.10   | 0.10 |
| RSC   | 0.66       | 0.23 | 0.11 | 0.00   | 0.00 |

Table 8: Category selectivity of ROI attention for subject 2.

|       | background | face | body | animal | food |
|-------|------------|------|------|--------|------|
| EBA   | 0.03       | 0.25 | 0.72 | 0.00   | 0.00 |
| FFA-1 | 0.05       | 0.57 | 0.19 | 0.18   | 0.00 |
| FFA-2 | 0.05       | 0.53 | 0.27 | 0.15   | 0.00 |
| OPA   | 0.74       | 0.15 | 0.06 | 0.05   | 0.00 |
| PPA   | 0.78       | 0.11 | 0.08 | 0.03   | 0.00 |
| RSC   | 0.71       | 0.19 | 0.09 | 0.00   | 0.00 |

Table 9: Category selectivity of ROI attention for subject 5.

|       | background | face | body | animal | food |
|-------|------------|------|------|--------|------|
| EBA   | 0.29       | 0.26 | 0.40 | 0.05   | 0.00 |
| FFA-1 | 0.12       | 0.67 | 0.13 | 0.08   | 0.00 |
| FFA-2 | 0.10       | 0.59 | 0.28 | 0.03   | 0.00 |
| OPA   | 0.31       | 0.33 | 0.20 | 0.16   | 0.00 |
| PPA   | 0.36       | 0.29 | 0.20 | 0.15   | 0.00 |
| RSC   | 0.08       | 0.37 | 0.50 | 0.00   | 0.06 |

Table 10: Category selectivity of ROI attention for subject 7.

|       | background | face | body | animal | food |
|-------|-----------|------|------|--------|------|
| EBA   | 0.38 | 0.05 | 0.43 | 0.09 | 0.05 |
| FFA-1 | 0.00 | 0.88 | 0.02 | 0.10 | 0.00 |
| FFA-2 | 0.17 | 0.20 | 0.38 | 0.20 | 0.05 |
| OPA   | 0.40 | 0.18 | 0.14 | 0.28 | 0.00 |
| PPA   | 0.51 | 0.29 | 0.16 | 0.05 | 0.00 |
| RSC   | 0.56 | 0.25 | 0.17 | 0.02 | 0.00 |

## A.3 Analyzing learned ROI queries

We analyzed the representational similarity of learned ROI queries, and report the average cosine similarity between each pair of ROIs across 20 models trained using five different random seeds and four different DINOV2 backbone layers in Figures S5, S6, S7, S8. These figures show the visual and semantic similarity between the ROIs as reflected in the learned queries for the subjects. We observed that ROIs with shared category selectivity form clusters (faces, places, bodies, or words) in the similarity matrix, exhibiting greater representational similarity within each category type.

We also see a clear divide between categorical and non-categorical areas. Additionally, ROIs within the ventral early visual areas (V1v, V2v, V3v) are more similar to one another than to their dorsal counterparts (V1d, V2d, V3d), and vice versa (the checkerboard patterns), reflecting the anatomical and functional organization of the visual cortex, and that the attention will be mostly driven by spatial information.

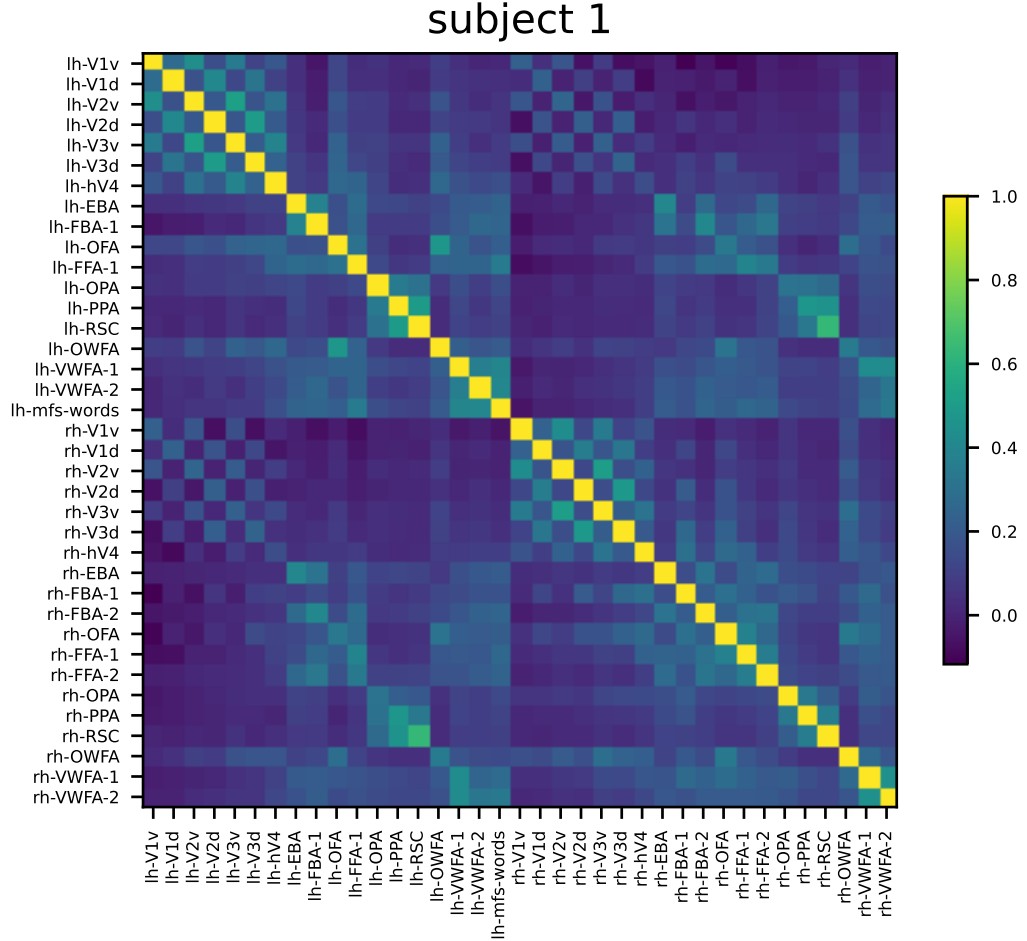

Figure S5: Cosine similarity between learned ROI queries for subject 1. Each entry in the matrix represents the average cosine similarity between the query for the ROI indicated by the row label and that indicated by the column label. ROIs from the left hemisphere are labeled with 'lh', and those from the right hemisphere with 'rh'. Results are averaged across 20 models, trained using five random seeds and four different backbone layers.

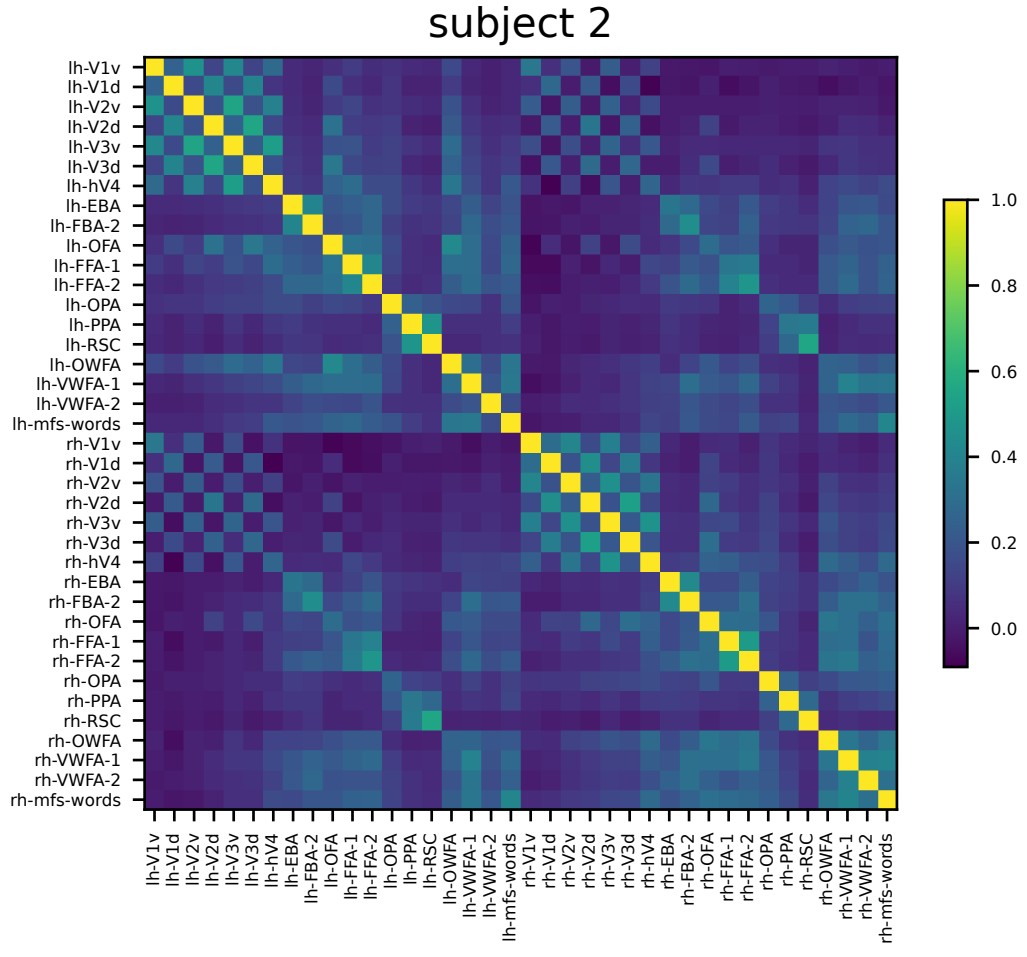

Figure S6: Cosine similarity between learned ROI queries for subject 2.

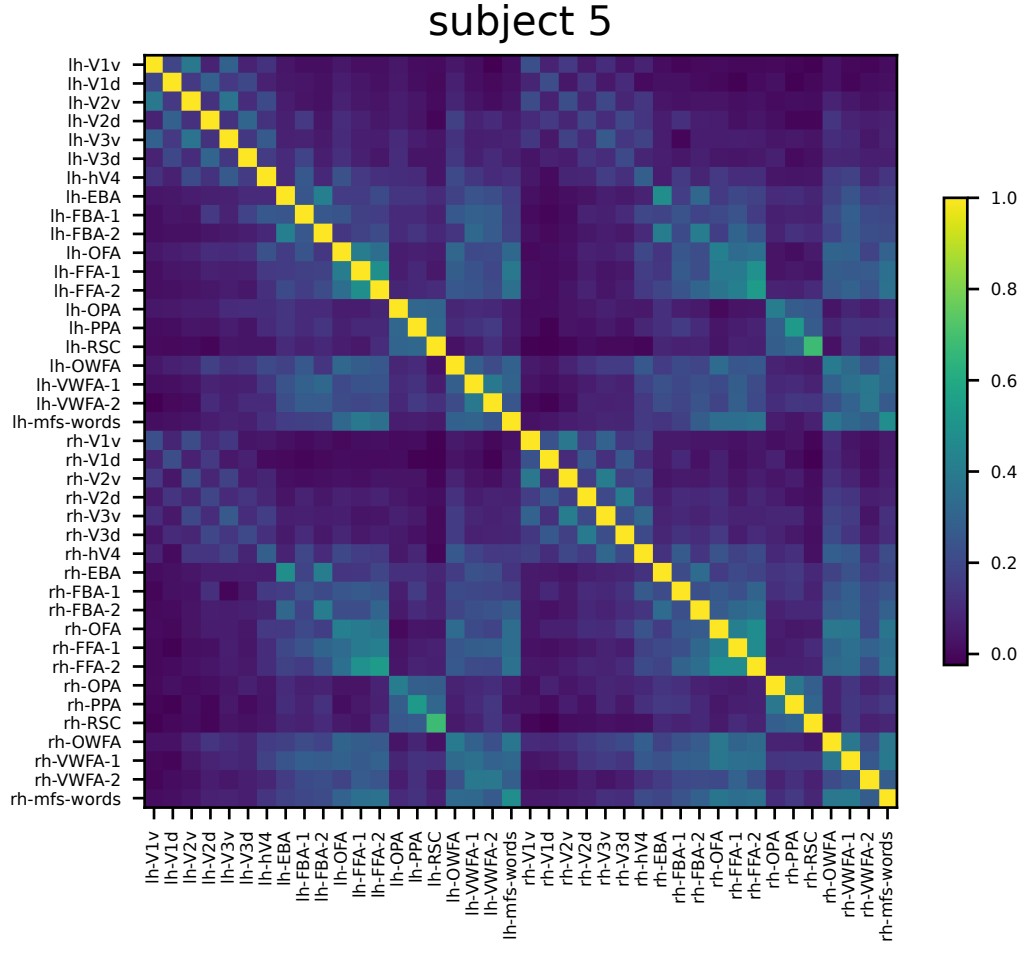

Figure S7: Cosine similarity between learned ROI queries for subject 5.

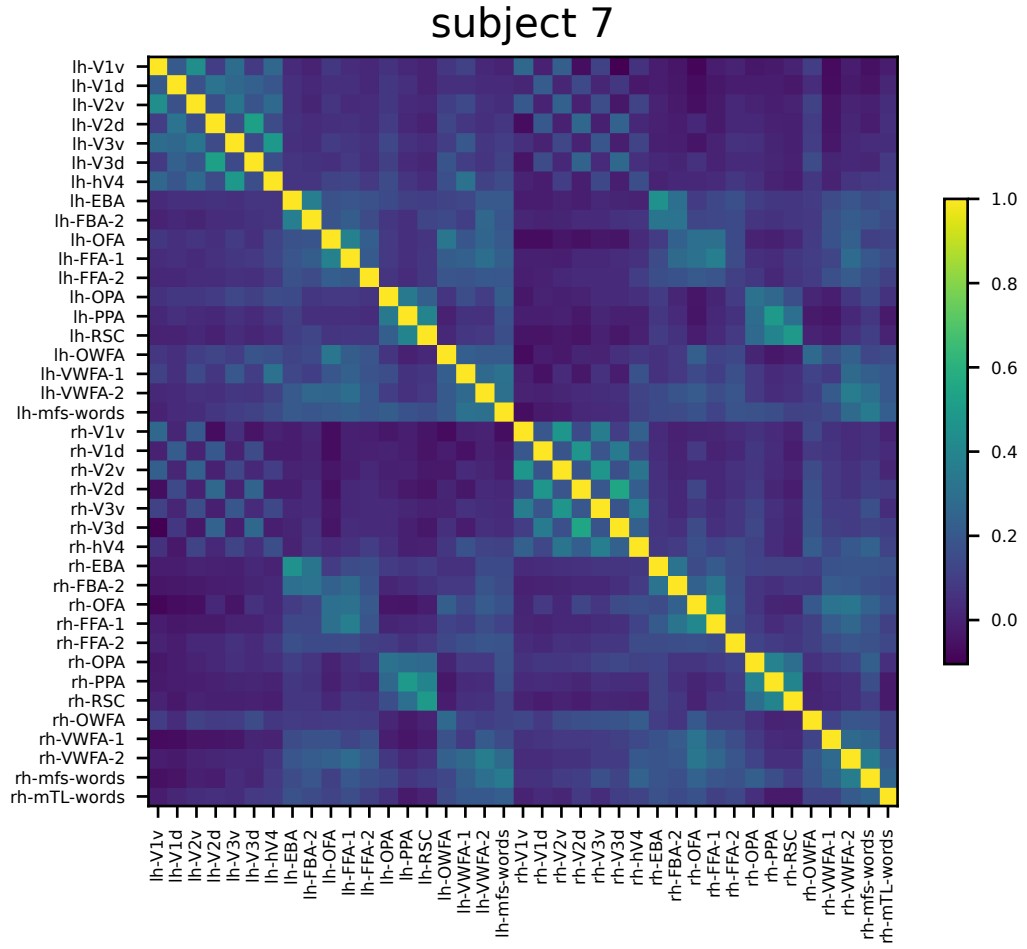

Figure S8: Cosine similarity between learned ROI queries for subject 7.

## A.4    Generating maximally activating images for ROIs

BrainDiVE [32] is a generative framework for synthesizing images predicted to activate specific regions of the human visual cortex. It guides the denoising steps of a diffusion model using gradients derived from a brain encoding model. Given the strong performance of our encoding model in predicting brain activity, we tested whether it could also effectively guide image generation within the BrainDiVE framework. We generated 200 images optimized to maximally activate the average predicted response of a specific ROI cluster, and display the top five in Figure S9, S10. The categories of the generated images are consistent with the reported category selectivity of each ROI cluster in the literature.

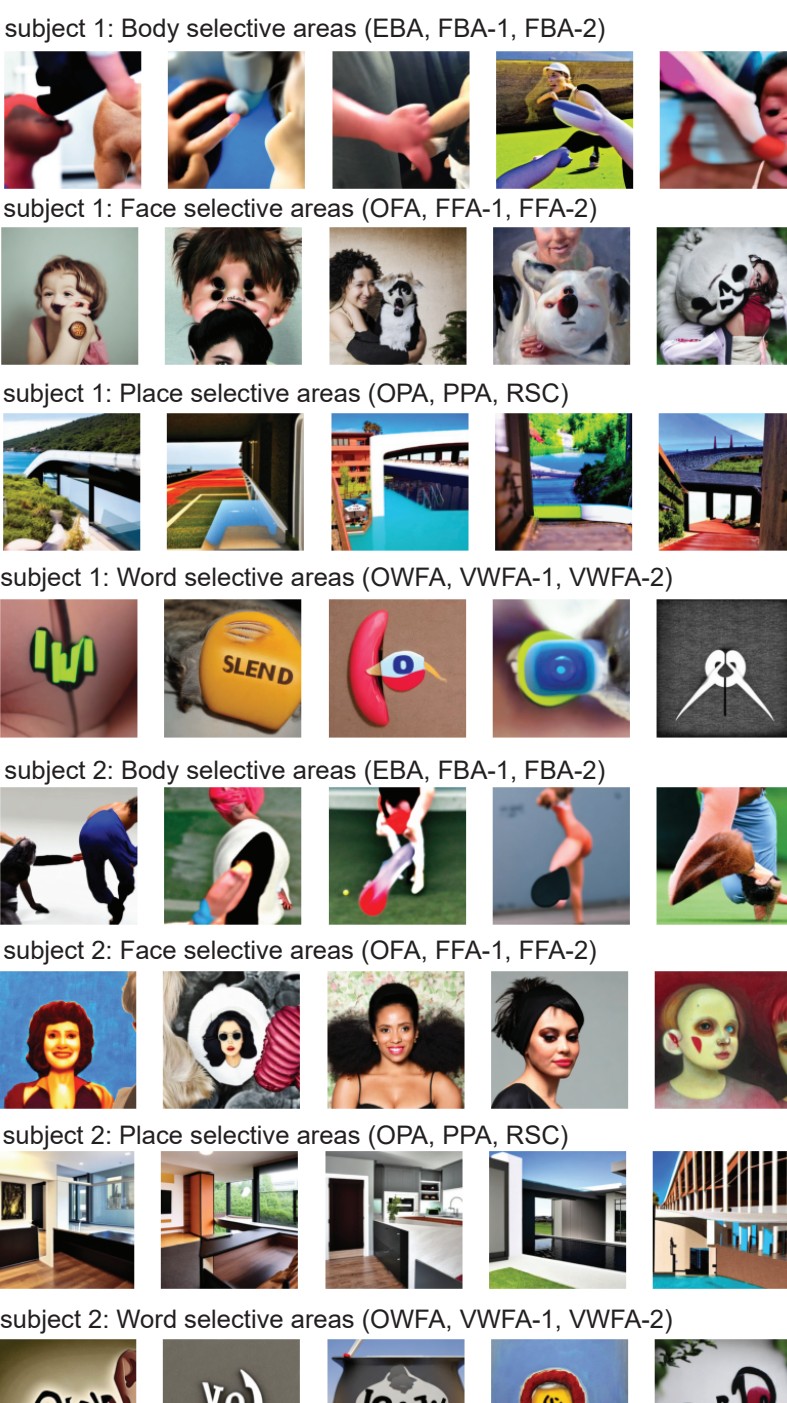

Figure S9: Images generated to maximally activate different ROI clusters for subjects 1 and 2. Using our encoding model within the BrainDiVE framework, we generated 200 images predicted to maximally activate a specific ROI cluster for a given subject (indicated by the row titles). For each cluster, we display the top five images with the highest predicted activation, as determined by our encoding model.

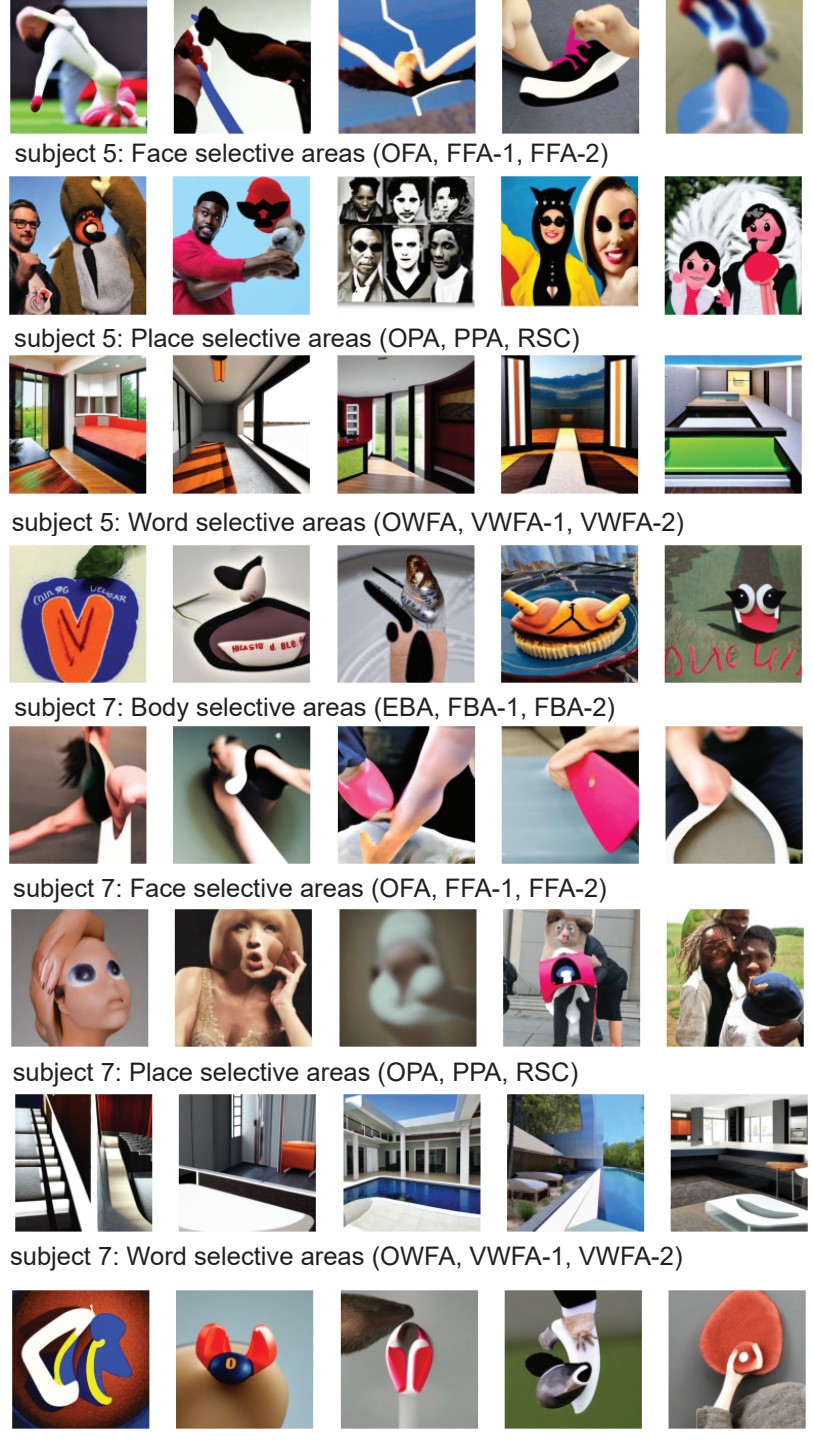

Figure S10: Images generated to maximally activate different ROI clusters for subjects 5 and 7

# B   Compute used

We used GPUs (NVIDIA L40s), memory, and storage resources from an internal cluster. Storage for the entire project totals roughly 3TB. Training the model used roughly 4,000 GPU hours. Running the remaining experiments used roughly 1,000 GPU hours. The full project required more compute than these estimates due to failed experiments, experiments not included in the paper, and model iteration.

