# OpenReview forum: "Transformer brain encoders explain human high-level visual responses"
_NeurIPS.cc/2025/Conference — NeurIPS 2025 spotlight_

### Official Review · Reviewer_RcVD · 2025-07-02

**Clarity:** 3
**Significance:** 3
**Originality:** 2
**Rating:** 5
**Confidence:** 4

**Summary:**

This work proposes a novel method for mapping activations in artificial neural networks to fRMI responses to visual stimuli. The method is based on an attention mechanism, where the model learns an input-independent query for each voxel (or ROI) which is compared to the token output of a backbone model using standard dot-product attention. The mechanism is characterized as an input-dependent receptive field that extracts only those features that are relevant to the voxel. The approach massively reduces the number of parameters of the readout module compared to other approaches and yields better predictive performance.

**Questions:**

1. Are both values and keys in the decoder architecture equal to the output tensor of the backbone? Fig. 1B suggests to me that key and value tensors are extracted from the backbone, but then I wouldn’t know how experiments were run on Resnet.
2. There are quite a few typos left in the text, eg.lines 45, 48, 135, 182, 235, 237, 238, 249, 259, 266

**Ethical Concerns:**

["NO or VERY MINOR ethics concerns only"]

**Final Justification:**

The thorough discussion of the questions I've raised has reinforced my position that this manuscript is suitable for publication. I've increased my score to 'Accept'.

**Limitations:**

yes

**Paper Formatting Concerns:**

I have no formatting concerns.

**Quality:**

3

**Strengths And Weaknesses:**

**Strengths**

1. The proposed method is very intuitive and easy to adapt for other researchers in the field. Therefore, I believe the paper has high relevance for the field.
2. Predictive accuracy is higher than for the other approaches described in the manuscript.
3. Qualitative attention maps yield results that are consistent with previously established tuning properties.
4. The paper is well-structured and well-written.

**Weaknesses**

1. The algorithmic novelty is somewhat limited, as the introduction of the attention mechanism is the only novel contribution over previous work.
2. Arguments are made that the complexity of the model does not take away from the correspondence between model activations and voxel responses (4.2). I would say that the attention mechanism is not over complex at all, but there is a lack of discussion on the fully connected layer between attention and linear map, which destroys the linearity. Is it necessary to achieve good performance?
3. Since there is 10-fold cross validation yielding 10 models, which are averaged, it is possible that there is an ensemble effect that contributes to the good performance. As far as I understand, competing models are not ensembled.
4. In my opinion, the paper misses two very important competing models: First, one based on the [cls] token representations which also extracts information from semantically salient regions similar to the proposed method. Second, one often reduces the dimensionality of the features in an unsupervised fashion. It would be good to compare the model to one based on PCA + Ridge.
5. Since the algorithmic advancement is on the smaller side, it might be desirable to include multiple datasets or data modalities in the analysis.

---

> ### Author Rebuttal · Authors · 2025-07-31
>
> We really appreciate the thoughtful review! Below we respond to all the comments and questions.
>
> > **1)** Algorithmic novelty of applying the attention mechanism
>
> As you and other reviewers pointed out, the application of the attention mechanism is an elegant solution to the problem we set out to solve that can be easily applied in many different setting and massively cuts down on the number of parameters while improving performance by taking into account the dynamics of brain computations. We believe this contribution is significant in many ways.
>
> We also want to highlight what other algorithmic explorations will be enabled by this approach. It connects the work to dynamic routing of information, understanding selectivity, and general connectivity of brain areas, among many other important topics in computational neuroscience and AI such as how neuronal circuits are reconfigured to dynamically route information for different purposes.
>
>
> > **2)** On the effect of linear mapping between attention and fMRI activation mapping
>
> As you pointed out, a fully connected feedforward operation is applied to all the tokens after the attention operation. In our experiments, removing this operation did not have any noticeable impact on the performance.
>
> The main non-linearity in our model is in the softmax applied to the scaled dot products to calculate the attention weights. We observed that removing this non-linearity had a significant impact on accuracy (~0.06 points decrease).
>
> We will discuss these points in the revision.
>
>
> > **3)** Ensemble effect
>
> You are right that there is some ensembling effect on the performance and for that exact reason we apply this cross-validation on all the models including all the baseline models. We agree otherwise it would not have been a fair comparison. We will make this clear in the revision.
>
> > **4)** Additional baselines ([cls] token and PCA + Ridge.)
>
>
> Excellent suggestions. We completely agree that these two baselines are important to include and thank the reviewer for pointing them out. We show their performance below in comparison to the Ridge regression and transformer attention-based encoding models.
>
> **Encoding accuracy using DINOv2 backbone**
>
> | Encoder                             | S1   | S2   | S5   | S7  |
> |-------------------------------------------|------|------|------|------|
> | Ridge regression                    | 0.56 | 0.52 | 0.50 | 0.37 |         |
> | Transformer (rois)                  | 0.60 | 0.56 | 0.56 | 0.42 |          |
> | CLS + regression         | 0.38 | 0.37 | 0.45 | 0.33 |           |
> | PCA + regression      | 0.52 | 0.47 | 0.46 | 0.34 |
>
>
> The CLS token takes a weighted average of all the important/salient image tokens to create a compact representation of the image. But mapping this token to all the vertices in the brain does not result in good performance. The big difference to note is that our model allows each ROI or vertex to dynamically route the tokens that have relevant content for that ROI or vertex so in a way each learn to create their own "CLS" token based on the content of the image and the ROI/vertex selectivity.
> The PCA based model performs better but still below the other two baselines. We will add these analyses to the revision.
>
>
> > **5)** Multiple datasets or data modalities in the analysis
>
> In this paper we performed our experiments on the NSD dataset, as this is the largest and most commonly used dataset in the field, making comparison to other works easier. But we have explored a range of different encoding models, feature backbones, and routing mechanisms in this work. We also provided results in section 4.4 (Fig. 6) for the text modality. We believe these results strongly support our claims in this paper and we leave testing on other datasets to future work.
>
>
> > **6)** Values and keys in the decoder architecture
>
> As shown in Fig. 1B, the value tensors are  extracted from the feature backbone. We then create positional embeddings based on the size of the feature maps (i.e. the number of tokens). The Keys are then set as values + positional embeddings. We follow the same exact process for all the models including ResNets.
>
>
> Thank you again for your comments! We hope our answers have helped clarify your questions, and we look forward to any additional discussion.

---

> > ### Comment · Reviewer_RcVD · 2025-08-01
> >
> > Thank you for the insightful discussion!
> >
> > 1) I completely agree that finding improved ways of connecting models to neural data is worthwhile. Still, I believe my point about the the somewhat limited algorithmic novelty stands and I have not been fully convinced otherwise. Of course, this criticism should be viewed in context and I am not arguing that the paper should be rejected on these grounds.
> > 2) I think that the non-linearity of the attention softmax is no issue at all, since the feature output is still a convex combination of the Transformer features. It is very interesting that the MLP layer connecting attention outputs and linear readout is not important for performance - in that case, I believe removing this layer could aid in terms of simplicity/interpretability.
> > 3) Thanks for the clarification!
> > 4) Thank you for the additional experiments - I believe that these results further strengthen the manuscript.
> > 5) I would agree that the lack of other datasets is not a strong drawback and it is acceptable to me to leave it for future work.
> > 6) Thank you for clarifying!
> >
> > Overall, I think the author's response to my comments was convincing and I am happy to recommend this work for publication.

---

> > > ### Author Response · Authors · 2025-08-04
> > >
> > > We really appreciate your positive assessment and thank you again for your suggestions and comments!

---

### Official Review · Reviewer_EjBm · 2025-07-02

**Clarity:** 3
**Significance:** 3
**Originality:** 3
**Rating:** 5
**Confidence:** 5

**Summary:**

This paper presents a transformer‐based readout model for mapping DNN features to brain responses. It employs content‐dependent, attention-driven routing of retinotopic features to category-selective cortical areas. Using large-scale fMRI data (NSD), the authors show that their single-layer decoder outperforms both ridge regression and spatial-feature factorized baselines across vision and text modalities.

**Questions:**

The paper needs some neuroscientific grounding. It remains unclear which biological phenomenon the transformer readout captures. Citing studies on content-dependent RF shifts (e.g., attention-modulated RF dynamics) would strengthen the link to brain physiology.

Alternative attention baselines: A simpler attention-modulation module (without a full transformer block) could serve as a useful comparison to isolate the benefit of cross-attention versus generic feature reweighting. For eg see Khosla et al.

ROI-level granularity: Applying the same dynamic routing across all voxels in an ROI may obscure within-ROI variability—particularly in early visual areas with heterogeneous RFs. This might explain the limited gains of vertex-based queries in higher early visual regions.

The spatial-feature factorized model could benefit from sparsity or smoothness constraints on learned RFs; exploring such priors might narrow the performance gap.

It is unclear how the ensemble results relate to the rest of the paper. Is the main conclusion that ensembling across layers primarily helps early visual areas? Do the authors have an intuition for why that is the case?

Beyond qualitative examples, measuring whether attention peaks overlap with expected stimulus features (e.g., faces for FFA, left visual field for FBA) would substantiate interpretability claims.

Minor typos:

    Line 45: change “has shown that…” to “have shown that…”

    Line 249: correct “TO test this” to “to test this.”

Refs: Khosla, Meenakshi, et al. "Neural encoding with visual attention." Advances in Neural Information Processing Systems 33 (2020): 15942-15953.

**Ethical Concerns:**

["NO or VERY MINOR ethics concerns only"]

**Final Justification:**

The authors addressed all my concerns with new baselines

**Limitations:**

yes

**Quality:**

3

**Strengths And Weaknesses:**

- The idea is simple and elegant. Framing dynamic receptive fields as an attention routing problem is both intuitive and powerful.
- The manuscript is well organized, with intuitive figures/results and clear explanations.
- The method is pretty general (can be applied to transformer architectures across modalities or even CNNs) - and they demonstrate gains with both image and caption inputs to highlight the generality of the method.

---

> ### Author Rebuttal · Authors · 2025-07-31
>
> We thank the reviewer for the very thoughtful review. Please see below for our responses.
>
> > **1)** Neuroscientific grounding for flexible routing
>
> Thank you for this insight. A flexible routing mechanism would indeed be reflected in deviation from the classical RF characterization of responses, so content-dependent RF shifts provide some evidence for a more flexible mechanism. Another important connection is to studies of low-rank communication between brain areas which can also be modulated by attention. We will add the following references to acknowledge the important links to the literature and would appreciate any other studies you can suggest.
>
>
> Ramalingam, N., McManus, J. N., Li, W., & Gilbert, C. D. (2013). Top-down modulation of lateral interactions in visual cortex. Journal of Neuroscience, 33(5), 1773-1789.
>
> David, S. V., Fritz, J. B., & Shamma, S. A. (2008). Task reward structure shapes receptive field modulation in primary auditory cortex. Nature Neuroscience, 11(6), 639–646.
>
> Gilbert, C. D., & Li, W. (2013). Top-down influences on visual processing. Nature Reviews Neuroscience, 14(5), 350–363.
>
> Womelsdorf, T., & Everling, S. (2015). Long-range attention networks: Circuit motifs underlying endogenously controlled stimulus selection. Trends in Neurosciences, 38(11), 682–700.
>
> João D Semedo, Amin Zandvakili, Christian K Machens, Byron M Yu, and Adam Kohn. Cortical areas
> interact through a communication subspace. Neuron, 102(1):249–259, 2019.
>
> > **2)** Alternative attention baseline
>
> Thanks for bringing the Khosla et al. (2020) study to our attention. It it very relevant and we agree the performance on this baseline is very informative.
>
> To implement this baseline, we used DeepGaze (Kummerer et al, 2022), a state the art saliency model, to generate bottom-up saliency maps for all the images in our dataset. We then resized the maps to $31*31$ and used the resulting attention values (weights) to combine the token representations to create a single token. A linear regression was then trained to map these compressed representations to vertex activation.
>
>
> **Encoding accuracy using DINOv2 backbone**
>
> | Encoder                             | S1   | S2   | S5   | S7  |
> |-------------------------------------------|------|------|------|------|
> | Saliency based integration                    | 0.38 | 0.37 | 0.44 | 0.32 |         |
>
> The table above shows the encoding accuracy for this model across the four subjects. As you can see from our response to reviewer RcVD under **additional baselines**, the encoding accuracy of this model is comparable to the CLS+regression model. This is not surprising as the CLS token does take a weighted average of all the other tokens that are important for classification and one would expect the important tokens for classification would also have high bottom-up activation. We will discuss the saliency-based baseline in the revision as it will indeed "serve as a useful comparison to isolate the benefit of cross-attention versus generic feature reweighting."
>
>
>
> > **3)** ROI-level granularity: Applying the same dynamic routing across all voxels in an ROI may obscure within-ROI variability—particularly in early visual areas with heterogeneous RFs. This might explain the limited gains of vertex-based queries in higher early visual regions.
>
> Exactly matches our intuition as well. We will make this point clear in the revision.
>
>
> > **4)** Ways to improve the the spatial-feature factorized baseline model
>
> Thank you for these suggestions, we did explore applying sparsity on the maps and did not see a noticeable difference in the accuracy, but we agree smoothness constraints are worth exploring as well.
>
>
> > **5)** Ensembling across layers results
>
>
> The ensembling results show how the encoding accuracy would improve by taking into account features from earlier layers of the feature backbone. And since those features are coming from earlier layers, and we observe that the improvement are mostly in earlier visual areas, this shows further correspondence between brain areas and feature layers. It also highlights our method being suitable for ensembling as it is desired in many applications.
>
>
> > **6)** Quantitative examination of attention maps to substantiate interpretability claims.
>
> Excellent suggestion. We have done experiments now quantifying this process. We classified each pixel of the test set images into five categories using YOLOv5 (Ultralytics., 2020) and YOLOv8-face ( Dhar, A., 2023): background, face, body, animal and food. For each ROI, we calculated and resized the attention maps to $434 * 434$ for these images, and report the categories of the$~2k$ pixels with top attention values.  We found that the category selectivity is consistent with ROI labels, with EBA most selective for body, FFA most selective for face, and OPA/PPA/RSC most selective for background.
>
> Below shows result for subject 1, we have observed similar results with subject 2, 5, 7 and will include them in the revision.
>
>
> |       | background | face | body | animal | food |
> |-------|------------|------|------|--------|------|
> | EBA | 0.03        | 0.36  | 0.61 | 0.00   | 0.00 |
> | FFA-1  |  0.00    | 0.79  | 0.16 | 0.05   | 0.00 |
> | FFA-2 | 0.00      | 0.83  | 0.17 | 0.00   | 0.00 |
> | OPA | 0.54        | 0.12  | 0.14 | 0.05   | 0.15 |
> | PPA | 0.44        | 0.25  | 0.11 |  0.10  | 0.10 |
> | RSC | 0.66        | 0.23  | 0.11 | 0.00   | 0.00 |
>
>
>
> Thank you again for your great suggestions. Please let us know if you have any additional questions or comments!

---

> > ### Author Response · Authors · 2025-08-07
> >
> > We hope our response has been helpful! We would be more than happy to discuss further.
> >
> > Thank you!

---

> > > ### Comment · Reviewer_EjBm · 2025-08-08
> > >
> > > I thank the authors for addressing all my concerns

---

> > > > ### Author Response · Authors · 2025-08-09
> > > >
> > > > We really appreciate your positive assessment of our work and thank you again for your detailed comments and feedback!

---

### Official Review · Reviewer_UAkU · 2025-07-03

**Clarity:** 4
**Significance:** 3
**Originality:** 3
**Rating:** 5
**Confidence:** 4

**Summary:**

This paper introduces a new framework with the attention mechanism from a the transformer unit to model feature routing from images to various regions of the visual cortex. It shows that the transformer based framework performs better and is more interpretable than the ridge based method for understanding visual representation in the brain.

**Questions:**

For Figure 2b ,2c and 3b,  showing all subjects average (instead of just subject 1) with errors bar will help readers get a sense of the improvement of performances with respect to individual variability.

**Ethical Concerns:**

["NO or VERY MINOR ethics concerns only"]

**Final Justification:**

Overall solid paper with a relatively high impact method. No unresolved issues.

**Limitations:**

Yes

**Quality:**

3

**Strengths And Weaknesses:**

Strengths:
- The framework proposed in the paper is novel.
- The framework itself is simple, yet very effective. The improvement of encoding accuracy is encouraging and this method largely fills in the gap where most encoding models ignore receptive field of the respective brain regions they are trained to model.
- I see the method widely applicable to most studies with encoding model with minimal extra computes/training time.
- The writing is very clear and the paper is organized well.

Weakness:
- To demonstrate the point that the attention module is "inherently more interpretable" I would expect the authors discover new scientific insight with this method. Attention map visualization from V2d, OFA and FBA shows the method indeed works. I am wondering whether the authors have tried visualizing less understood areas? Besides that, attention map of vertices along certain axis could be potentially very interesting maybe we could see smooth gradient of feature coding both semantically and spatially.

---

> ### Author Rebuttal · Authors · 2025-07-31
>
> We thank the reviewer for their thoughtful comments and feedback! Please see below for our responses.
>
>
> > **1)** Interpreting attention maps
>
> In this work, we focused on the NSD-general subset of areas in the brain, comprising mostly of visually responsive vertices in the back of the brain (Fig.1 A). These vertices are mostly labeled with known categorical selectivity. However, in parallel work, we have extended our model to predict the activity for the parcels across the whole brain and we have observed the effectiveness of our method in labeling unknown regions even beyond the traditional visual cortex (e.g. ROI for tool use).
>
> However, to strengthen our claims on attention effectiveness, we have done experiments now to quantitatively examine the maps. We classified each pixel of the test set images into five categories using YOLOv5 (Ultralytics., 2020) and YOLOv8-face ( Dhar, A., 2023): background, face, body, animal and food. For each ROI, we calculated and resized the attention maps to $434 * 434$ for these images, and report the categories of the$~2k$ pixels with top attention values.  We found that the category selectivity is consistent with ROI labels, with EBA most selective for body, FFA most selective for face, and OPA/PPA/RSC most selective for background.
>
> Below shows result for subject 1, we have observed similar results with subject 2, 5, 7 and will include them in the revision.
>
> |       | background | face | body | animal | food |
> |-------|------------|------|------|--------|------|
> | EBA | 0.03        | 0.36  | 0.61 | 0.00   | 0.00 |
> | FFA-1  |  0.00    | 0.79  | 0.16 | 0.05   | 0.00 |
> | FFA-2 | 0.00      | 0.83  | 0.17 | 0.00   | 0.00 |
> | OPA | 0.54        | 0.12  | 0.14 | 0.05   | 0.15 |
> | PPA | 0.44        | 0.25  | 0.11 |  0.10  | 0.10 |
> | RSC | 0.66        | 0.23  | 0.11 | 0.00   | 0.00 |
>
>
>
> > **2)** Smooth gradient of feature coding both semantically and spatially revealed by attention maps
>
> A unique aspect of our approach is that it provides learned queries that determine the type of information that would be routed to each ROI. We can therefore study the similarity of the query embeddings to study attention behavior (more similar query embeds lead to more similar attention maps between ROIs). Supplementary figures S5 - S8 (Sec A.2) show the cosine similarity between all these learned ROI queries. These similarities are also highly correlated with functional connectivity between these areas.
>
> As the reviewer points out, the graded attention routing signal emerges along the hierarchy with areas becoming less spatial and more categorical. In early visual areas a checkerboard pattern emerges, where V1v in left hemisphere routes very similar information as the V2v in the same hemisphere (ventral ROIs more similar to one another than to their dorsal counterparts). Meanwhile, attention maps for higher visual ROIs are more semantically selective and form categorically selective clusters. We will add more discussion of this graded process in the revision.
>
> > **3)** For Figure 2b ,2c and 3b, showing all subjects average (instead of just subject 1) with errors bar will help readers get a sense of the improvement of performances with respect to individual variability.
>
> Thank you for the excellent suggestion, we will include this figure in the revision.
>
> We hope that our responses have addressed your comments and we look forward to any additional questions!

---

> > ### Comment · Reviewer_UAkU · 2025-08-05
> > **Thank you for the response.**
> >
> > Thank you for the response. I have no further questions or concerns.

---

> > > ### Author Response · Authors · 2025-08-06
> > >
> > > Thank you for the positive evaluation of our work!

---

### Official Review · Reviewer_dqys · 2025-07-03

**Clarity:** 4
**Significance:** 3
**Originality:** 3
**Rating:** 5
**Confidence:** 4

**Summary:**

This paper proposes and investigates the use of transformer encoder-decoder models in predicting fMRI responses of subjects seeing images from the Natural Scene Dataset. These transformer models outperform linear mapping/ridge-regression based on transformer-features for several different "feature backbones." The authors also find higher performance for vertex-based routing vs. ROI-based routing, interpretable attention maps, and some generalization to text (BERT backbone).

**Questions:**

1) Is there a simple way to understand the significance of an encoding accuracy between for example 0.6 and 0.8, as seen in Figure 2C? For example, one way to get at this qualitatively could be by evaluating quality of decoding based on the given encoding model. It would also be helpful to state what the best explained variance one might reasonably expect on this data (e.g., due to noise).

2) Can the authors provide a definition of vertex (ideally upon first use in Section 4.1)?

3) The authors write, "Our approach proposes a mechanistic model of how visual information from retinotopic maps can be routed based on the relevance of the input content to different category-selective regions." Could they comment on how this model could be related to experiments in systems neuroscience, or models in computational neuroscience such as dynamic routing?

3) How does the performance of transformer-based methods scale with the amount of data used?

4) How much of an impact do parameters like temperature impact encoding accuracy?

Minor comments:
1) possible missing ref on Line 134
2) For Figures 3b/4c, though the caption makes clearer that the dash should be read as "minus", it would be nice to make this clearer from the subtitle itself.
3) Is it possible to add some context/colorbar for the attention maps in Figure 5?

**Ethical Concerns:**

["NO or VERY MINOR ethics concerns only"]

**Final Justification:**

I believe that this paper has a strong impact for an active research area in computational neuroscience, and the results adequately support the proposed approach. The reviewers have adequately addressed my questions and their followup experiments were insightful.

**Limitations:**

Yes

**Quality:**

3

**Strengths And Weaknesses:**

I'm happy to recommend this work for acceptance. The authors are working on a highly relevant problem in neuroscience and propose a simple yet powerful alternative to the "standard" linear encoding approaches to predicting brain activity. The main claims of the paper are supported by experiments from multiple encoding "backbones". The explanations and motivations for different design choices is also well explained in the paper.

I think the work could benefit further by explaining how to translate higher predictive performance into qualitative insights for particular ROIs or voxels. I recognize the supplement (Figure S9) is making an attempt at this, but I think it would also be helpful to give insights into why the transformer is performing better than linear regression in certain cases. Another idea is to go further into detail into the improved interpretability relative to other approaches.

---

> ### Author Rebuttal · Authors · 2025-07-31
>
> We thank the reviewer for their very thoughtful comments and feedback! Please see below for our responses.
>
> > **1)** Insights from better performance
>
> We agree that it is very important to translate better accuracy into insights about brain computation. The big insight in our work is that the encoding performance reflects whether the routing in the model is capturing the routing of information in the brain (i.e. receptive field dynamics). For the different results we highlighted this insight to give better intuitions but will do so more in the revision (e.g. an insight highlighted by reviewer EjBm: "ROI-level granularity: Applying the same dynamic routing across all voxels in an ROI may obscure within-ROI variability—particularly in early visual areas with heterogeneous RFs. This might explain the limited gains of vertex-based queries in higher early visual regions.").
>
> > **2)** Improved interpretability relative to other approaches.
>
> Thank you for this suggestion. We showed attention maps in Fig. 5 and discussed how the attention behavior can be interpreted to reveal the selectivity of an ROI.
>
> We have now run experiments quantifying this process. We classified each pixel of the test set images into five categories using YOLOv5 (Ultralytics., 2020) and YOLOv8-face ( Dhar, A., 2023): background, face, body, animal and food. For each ROI, we calculated and resized the attention maps to $434 * 434$ for these images, and report the categories of the$~2k$ pixels with top attention values.  We found that the category selectivity is consistent with ROI labels, with EBA most selective for body, FFA most selective for face, and OPA/PPA/RSC most selective for background.
>
> Below shows result for subject 1, we have observed similar results with subject 2, 5, 7 and will include them in the revision.
>
> |       | background | face | body | animal | food |
> |-------|------------|------|------|--------|------|
> | EBA | 0.03        | 0.36  | 0.61 | 0.00   | 0.00 |
> | FFA-1  |  0.00    | 0.79  | 0.16 | 0.05   | 0.00 |
> | FFA-2 | 0.00      | 0.83  | 0.17 | 0.00   | 0.00 |
> | OPA | 0.54        | 0.12  | 0.14 | 0.05   | 0.15 |
> | PPA | 0.44        | 0.25  | 0.11 |  0.10  | 0.10 |
> | RSC | 0.66        | 0.23  | 0.11 | 0.00   | 0.00 |
>
>
>
> A unique aspect of our approach is that it provides learned queries that determine the type of information that would be routed to each ROI. We can therefore study the similarity of the query embeddings to study attention behavior (more similar query embeds lead to more similar attention maps between ROIs). Supplementary figures S5 - S8 (Sec A.2) show the cosine similarity between all of these learned ROI queries.
>
> A graded attention routing signal emerges along the hierarchy with areas becoming less spatial and more categorical. In early visual areas a checkerboard pattern emerges, where V1v in left hemisphere routes very similar information as the V2v in the same hemisphere (ventral ROIs more similar to one another than to their dorsal counterparts). Meanwhile, attention maps for higher visual ROIs are more semantically selective and form categorically selective clusters.
>
> We believe these two analyses strengthen our claims on improved interpretability of our approach compared to alternatives. And we will highlight them more in the revision.
>
> > **3)**  Understanding the significance of an encoding accuracy
>
> All the results presented in our paper are based on encoding accuracy normalized by noise ceiling, indicating the portion of the variance that is explained. So the ceiling is always at 1 (that's as good as we expect any model to perform).
>
>
> > **4)** Define vertex early
>
> Sure, we will add the below definition and ref in section 4.1.
>
> "In fMRI (functional magnetic resonance imaging), a vertex typically refers to a point on the surface mesh of the cortex used in surface-based analysis. It is analogous to a "voxel" (volumetric pixel) in volumetric fMRI data, but defined in the 2D cortical surface space."
>
> Fischl, B. (2012). FreeSurfer. NeuroImage, 62(2), 774–781. https://doi.org/10.1016/j.neuroimage.2012.01.021
>
> > **5)** Relation to models in comp neuro models
>
> Relational processing in the visual and other perceptual systems is thought to be implemented through lateral and large-range connectivity in the primate brain. However, how brain regions flexibly access the results of each others inferential computations is not well understood. A content-specific selective access mechanism as provided by transformer attention is probably also important for the primate brain. Our work shows that including such a mechanism in a brain-computational model improves the model's ability to predict neural responses. Although the particular mechanism for flexible routing implemented in the primate brain is yet to be discovered, our work contributes to the evidence that the human brain employs some such mechanism. We will add a discussion on this in the revision and plan to use the following references to acknowledge the important links to the literature.
>
> Ramalingam, N., McManus, J. N., Li, W., & Gilbert, C. D. (2013). Top-down modulation of lateral interactions in visual cortex. Journal of Neuroscience, 33(5), 1773-1789.
>
> David, S. V., Fritz, J. B., & Shamma, S. A. (2008). Task reward structure shapes receptive field modulation in primary auditory cortex. Nature Neuroscience, 11(6), 639–646.
>
> Gilbert, C. D., & Li, W. (2013). Top-down influences on visual processing. Nature Reviews Neuroscience, 14(5), 350–363.
>
> Womelsdorf, T., & Everling, S. (2015). Long-range attention networks: Circuit motifs underlying endogenously controlled stimulus selection. Trends in Neurosciences, 38(11), 682–700.
>
> João D Semedo, Amin Zandvakili, Christian K Machens, Byron M Yu, and Adam Kohn. Cortical areas interact through a communication subspace. Neuron, 102(1):249–259, 2019.
>
>
> > **6)** The effect of training data size on the performance
>
> Excellent suggestion. Our models were all trained with the training split of size 8800 images. We now experiment with training the network with different fractions of the total number of training images (550, 1100, 2200, and 4400) and display the results below.
>
> **Encoding accuracy using DINOv2 backbone**
>
> | Encoder                                               | Training set size   | S1   | S2   | S5   | S7  |
> |------------------------------------------------|------|------|------|------|------|
> | Ridge regression        |8800      | 0.56 | 0.52 | 0.50 | 0.37 |
> | Transformer (rois) | 550                         | 0.41 | 0.45 | 0.45 | 0.34 |
> | Transformer (rois) |1100                       | 0.46 | 0.48 | 0.49 | 0.36 |
> | Transformer (rois) | 2200                       | 0.51 | 0.52 | 0.51 | 0.39 |
> | Transformer (rois) | 4400                       | 0.56 | 0.54 | 0.54 | 0.40 |
> | Transformer (rois) | 8800                       | 0.60 | 0.56 | 0.56 | 0.42 |
>
>
> It is very encouraging to see that our approach can be trained using with as little as a few hundred samples making it suitable for smaller scale experiments as well. Also encouraging is to see that our model can achieve accuracy on par with the baseline models with a fraction of the training set.
>
> > **7)** Temperature parameter
>
> In all of our experiments, we used the most common choice of the scaling value, which is $1/sqrt(d)$ and with d = 768 in our model (so ~0.036). The motivation was that this parameter in many implementations of attention layer is not easily accessible so keeping the default scaling value and optimizing other hyperparameters makes it easier to use for other researchers.
>
> However, to address reviewer's question, we did an experiment manipulating the scaling value from 0.001 to 1.0. For values near the default value (0.01 to 0.1) the encoding accuracy was very similar to the one from the default value, but moving to more extreme values (e.g. 1.0) led to lower performance. Although not conclusive, it appears to us that the default scaling value that has worked well across many domains seems to also be near optimal in this task.
>
>
> > **8)** MISC
>
> We will add the missing ref, make the "minus" sign more clear in the figure, and will add color bar to the attention map. Thank you for pointing those out. And we hope our responses have addressed your comments, and look forward to any additional discussion!

---

> > ### Comment · Reviewer_dqys · 2025-08-04
> > **Reply to rebuttal**
> >
> > Thank you very much for this response! I greatly appreciate the authors' detailed, clear, and informative explanations. The authors have adequately addressed my technical questions.
> >
> > The additional experiments reported (parts 2, 6, and 7) strengthen and support the claims made in the paper. It's indeed interesting to see how selectivity (2) relates to the brain regions one might intuitively expect, and I agree with the authors on (6) regarding data efficiency.
> >
> > The authors' plan to include a discussion in relation to (5) with proposed references sounds great, and I would look forward to reading it. Congrats again on a nice paper, and thank you for the time and engagement you've put into this review process!

---

> > > ### Author Response · Authors · 2025-08-04
> > >
> > > We really appreciate the positive assessment you've given to our work! and we want to thank you again for your suggestions and comments.

---

### Note · Authors · 2025-08-14

We are deeply grateful for the reviewers' detailed and constructive comments, and appreciate the positive evaluation of our work ("**a simple yet powerful alternative to the "standard" linear encoding**" (dqys); "**widely applicable to most studies**" (UAkU); "**elegant**...**both intuitive and powerful**" (EjBm). "**method is very intuitive and easy to adapt for  other researchers in the field**" (RcVD)).




### 1. General clarifications
* We use the transformer attention mechanism to model the flexible routing of information in the brain.

* By taking into account the dynamics of brain computations, our approach achieves state of the art performance in predicting brain activity while massively cutting down on the number of parameters.

* Our method can be easily applied in many different settings even with relatively small number of datapoints.

### 2. New results

* For Reviewer dqys, we tested how the performance of our encoding model changes based on the number of training samples.

* For reviewer UAkU, we added quantitative evaluation of the transformer attention maps.

* For Reviewer EjBm, we included an additional baseline using bottom-up attention to combine feature tokens.

* For Reviewer RcVD, we added two new baselines, using alternative methods of compressing patch representation (using CLS token and PCA).


### 3. Writing and Scope


We elaborate more on the neuroscientific grounding of our approach in the revision and connect the work to studies on how neuronal circuits are reconfigured to dynamically route information for different purposes.

We thank all the reviewers for suggestions regarding our writing and clarity. We believe they will significantly improve the communication of our work.

Best,
Authors

---

### Decision · Program_Chairs · 2025-09-17

**Decision:**

Accept (spotlight)

**Comment:**

a) The authors proposes a new framework based on a transformer (attention-based architecture) to relate fMRI activity to ANN image features.

b) - The framework is novel, simple and effective and transferable to other similar applications.
- It can be used to identify attention maps in the brain
- Well written / easy to follow / intuitive figures.

c) Minor or resolve during the rebuttal period.

d) This paper is highly relevant to Neurips. All reviewers have recognized the high quality of the work both in substance and in form. The audience of this paper covers both the AI and the Neuroscience communities which at least justifies a spotlight (if not an oral).

e) The rebuttal period was helpful to resolve all the (mostly) minor concerns that the reviewers initially had about the paper improving their positive opinion towards acceptance.